# Single-cell transcriptomics identifies Keap1-Nrf2 regulated collective invasion in a *Drosophila* tumor model

**Deeptiman Chatterjee[1]\*[†], Caique Almeida Machado Costa[1], Xian-Feng Wang[1], Allison Jevitt[2‡], Yi-Chun Huang[1], Wu-Min Deng[1,2]\***

[1]Department of Biochemistry and Molecular Biology, Tulane University School of Medicine, Tulane Cancer Center, New Orleans, United States; [2]Department of Biological Science, Florida State University, Tallahassee, United States

**\*For correspondence:**
dchatterjee@tulane.edu (DC);
wdeng7@tulane.edu (W-MinD)

**Present address:** [†]Cold Spring Harbor Laboratory, Cold Spring Harbor, NY 11724, United States; [‡]Cancer and Cell Biology Department, Oklahoma Medical Research Foundation, Oklahoma City, United States

**Competing interest:** The authors declare that no competing interests exist.

**Abstract** Apicobasal cell polarity loss is a founding event in epithelial–mesenchymal transition and epithelial tumorigenesis, yet how pathological polarity loss links to plasticity remains largely unknown. To understand the mechanisms and mediators regulating plasticity upon polarity loss, we performed single-cell RNA sequencing of *Drosophila* ovaries, where inducing polarity-gene *l(2)gl*-knockdown (Lgl-KD) causes invasive multilayering of the follicular epithelia. Analyzing the integrated Lgl-KD and *wildtype* transcriptomes, we discovered the cells specific to the various discernible phenotypes and characterized the underlying gene expression. A genetic requirement of Keap1-Nrf2 signaling in promoting multilayer formation of Lgl-KD cells was further identified. Ectopic expression of Keap1 increased the volume of delaminated follicle cells that showed enhanced invasive behavior with significant changes to the cytoskeleton. Overall, our findings describe the comprehensive transcriptome of cells within the follicle cell tumor model at the single-cell resolution and identify a previously unappreciated link between Keap1-Nrf2 signaling and cell plasticity at early tumorigenesis.

## Editor's evaluation

This work demonstrates the power of single-cell omics and imaging analyses to identify cell types and factors playing a role in the disruption of polarity, a process relevant to epithelial cancers. The authors' claims are well supported by the data and analyses. Overall, this work is viewed as an important contribution to cell biologists who work on the epithelial morphogenesis or tumorigenesis.

## Introduction

Apical–basal cell polarity acts as a major gatekeeper against tumorigenesis in epithelial tissues (*Royer and Lu, 2011*). While its dysregulation is commonly associated with tumors (*Chatterjee and Deng, 2019*; *Rudrapatna et al., 2012*), cell polarity disruption is tightly controlled to also allow cellular movement during critical developmental processes such as gastrulation and wound healing (*Barriere et al., 2015*). This regulation of cell polarity enables the cells to undergo changes in plasticity, which is the ability of cells to change their phenotype in response to environmental factors without acquiring genetic mutations. Plasticity in these cells facilitates epithelial-to-mesenchymal transition (EMT) causing them to lose apical–basal polarity and weaken cell–cell adhesion with the purpose of promoting their movement beyond the confined space of the tissue (*Moreno-Bueno et al., 2008*; *Plygawko et al., 2020*). For these reasons, EMT is also associated with behaviors such as increased invasiveness and

**eLife digest** In the body, most cells exhibit some form of spatial asymmetry: the compartments within the cell are not evenly distributed, thereby allowing the cells to know whether a surface is on the 'outside' or the 'inside' of a tissue or organ. In the cells of epithelial tissues, which line most of the cavities and the organs in the body, this asymmetry is known as apical-basal polarity. Maintaining apical-basal polarity in epithelial cells is one of the main barriers that stops cancer cells from invading other tissues, which is the first step of metastasis, the process through which cancer cells leave their tissue of our origin and spread to distant locations in the body.

In the fruit fly *Drosophila melanogaster*, scientists have engineered cells in several tissues to stop producing the proteins that help establish apical-basal polarity, in an effort to study the earliest steps of tumor formation. Unfortunately, these experiments frequently lead to rampant metastasis, making it difficult to identify the earliest changes that make the tumor cells more likely to become invasive. Therefore, finding a tissue in which loss of apical-basal polarity does not cause aggressive cancer progression is necessary to address this gap in knowledge.

The epithelial cell layer lining the ovaries of fruit flies may be such a tissue. When these cells lose their apical-basal polarity, rather than becoming metastatic and spreading to distant organs, they interleave with each other, forming a tumorous growth that only invades into the neighboring compartment. Chatterjee et al. used this system to study individual invasive cells. They wanted to know whether the genes that these cells switch on and off are known to be involved in human cancers, and if so, which of them control the invasive behavior of tumor cells.

Chatterjee et al. determined that when cells in the fruit-fly ovary lost their polarity, they turned genes on and off in a pattern similar to that seen both in mammalian cancers and in tumors from other fly tissues. One of the notable changes they observed in the ovarian cells that lost apical-basal polarity was the activation of the Keap1/Nrf2 oxidative-stress signaling pathway, which normally protects cells from damage caused by excessive oxidation. In the ovarian cells, however, the activation of these genes also led to aggressive invasion of the collective tumor cells into the neighboring compartment.

Interestingly, this increase in invasiveness was characterized by polarized changes within the cells, specifically in the scaffolding that allows cells to keep their shape and move: the edge of the cells leading the invasion had greater levels of a protein called actin, which enables the cells to protrude into the neighboring compartments.

Chatterjee et al. have identified a new mechanism that impacts the migratory behavior of cells. Insights from their findings will pave the way for a better understanding of how and when this mechanism plays a role in metastasis.

metastatic migration of cancer cells (*Brabletz et al., 2018*; *Mani et al., 2008*). Recently emerging consensus in the field of EMT research draws attention to the presence of a continuum of metastable cells undergoing partial-EMT in cancer tissues, instead of one stable state of either epithelial or mesenchymal identity (*Grigore et al., 2016*; *Saxena et al., 2020*). A significant gap exists in our understanding of how these different metastable cell states are supported by the underlying gene expression and how it determines the overall behavior of the tumor cells.

Epithelial polarity is maintained by the mutually antagonizing, spatially restricted protein complexes at the apical and basolateral cytosolic domains of the cell that are frequently found disrupted in several cancers (*Elsum et al., 2012*; *Huang and Muthuswamy, 2010*; *Parker et al., 2014*). A causative link between polarity loss and tumorigenesis has since been established in the fruit fly *Drosophila melanogaster*, where the loss of basolateral polarity proteins – such as Scribble (Scrib), Discs large (Dlg), and Lethal giant larvae (l(2)gl or simply, Lgl) – combined with oncogenic Notch or Ras signaling, causes malignant tumorigenesis in several tissues (*Chatterjee and Deng, 2019*; *Enomoto et al., 2018*; *Papagiannouli and Mechler, 2004*; *Papagiannouli and Mechler, 2013*; *St Johnston and Ahringer, 2010*). While several advances have since been made in identifying genetic factors that regulate tumorigenesis in the wing-disc tumor model (*Atkins et al., 2016*; *Dillard et al., 2021*; *Doggett et al., 2015*; *Logeay et al., 2022*), our understanding of the mechanisms driving neoplastic remodeling of the tissue is lacking. This limitation is due to (1) the use of bulk tissue-based approaches that cannot resolve the cellular heterogeneity inherent to the growing tumor, and (2) rampant malignancy in the

wing-disc tumor model that obscures characterization of individual cell behavior. In a recent study, we drew attention to the phenotypic heterogeneity within multilayered follicle cells of *Drosophila* ovaries where Lgl-knockdown (Lgl-KD) is induced to cause polarity loss (*Jevitt et al., 2021*). Even in the presence of oncogenic Notch signaling, these multilayered cells displayed continued survival and sustained growth without causing long-distance metastasis when transplanted into the host's abdomen. This lack of oncogenic complexity provides an opportunity to study the behavior of discrete dysplastic cells upon polarity loss and determine how the underlying gene expression might regulate it.

The normally developing *Drosophila* follicular epithelium has been instrumental in our understanding of cellular plasticity that facilitates developmental EMT and collective migration of the anterior follicle cells and border cells (*Silver et al., 2001*). In 2019, we published a comprehensive single-cell transcriptomic atlas describing the gene expression signatures of individual $w^{1118}$ follicle cells, which also included the abovementioned cell types (*Jevitt et al., 2020*). In this study, we utilized the $w^{1118}$ follicle cell dataset as a template to isolate Lgl-KD cells that exhibit gene expression that is divergent from that of the normally developing (as in $w^{1118}$) follicle cells. We characterized the transcriptomic signatures of these abnormal cells and validated their association with the distinct phenotypes that are observed within the ovaries containing Lgl-KD follicle cells. The oxidative stress-responsive Keap1-Nrf2 signaling pathway was found to be enriched in the multilayered cells and unexpectedly, manipulating the expression of both Keap1 and Nrf2 was found to regulate the invasiveness of collective Lgl-KD cells with significant F-actin remodeling. Overall, our findings in this study describe the earliest transcriptomic changes within dysplastic cells of the follicular epithelia with polarity loss in its cells and identify novel regulatory pathways that determine plasticity in the invading tumor.

## Results

### Lgl silencing causes distinct phenotypic and transcriptomic changes in egg chambers

Using the pan follicle cell *traffic jam* (*tj*)-Gal4 driver with temperature-sensitive (TS) Gal80 (Gal80$^{TS}$) repressor to control transcriptional induction, follicle cell-specific RNAi-mediated silencing of Lgl (Lgl-KD) was induced in adult female flies for 72 hr, which resulted in the premature failure of oogenesis. This failure of egg development resulted in the accumulation of degenerated egg chambers within the epithelial sheath (*Figure 1A*, *Figure 1—figure supplement 1A*). Additionally, phospho-Histone 3 (pH3) accumulation was observed in the follicle cells at the egg chamber termini during mid-oogenesis, when normal follicle cells of the experimental control and Lgl-KD follicle cells in the lateral epithelia cease to undergo mitosis (*Figure 1A*). Cross-section of these egg chambers revealed significant multilayering of the epithelia, where cells exhibited decreased expression of the differentiated follicle cell marker Hindsight (Hnt) (*Sun and Deng, 2007*) and increased expression of the immature cell marker Cut (*López-Schier and St Johnston, 2001*; *Sun and Deng, 2005*), suggesting changes to the expected cell fate (*Figure 1—figure supplement 1B and C*). To determine how Lgl loss of function affects epithelial cell plasticity, we stained them with common epithelial cell markers Shotgun or Shg (also known as DE-Cad, the *Drosophila* homolog of E-cadherin) and its binding partner Armadillo or Arm (*Drosophila* homolog of α-catenin). Both Shg and Arm staining displayed gradually decreasing enrichment at the multilayered cell junctions along the apical–basal axis, with the basal-most layer expressing Shg at levels comparable to that observed in the monolayered cells of the experimental control egg chambers (*Figure 1B*). When the intensities of F-actin (Phalloidin) and Shg were measured along the apical–basal axis of the invasive multilayer shown in *Figure 1B*, significant decrease in Shg enrichment was observed along with mildly elevated F-actin at the apical-most tip of the invading group of cells (*Figure 1C*). Apically invasive movement was also detected in the single-cell derived, positively marked (GFP+) $lgl^{RNAi}$ MARCM clones, as well as groups of mitotic clones derived from negatively marked (GFP-) homozygous $lgl^4$ mutant follicle cells (*Figure 1—figure supplement 1D*). Quantifying our observations across five independent experiments, we found that multilayering at midoogenesis was the most prevalent (83.25%; n = 280) large-scale phenotype observed across the different stages of oogenesis in $tj^{TS}$>$lgl^{RNAi}$ ovarioles after 72 hr of Lgl-knockdown (72h-Lgl-KD) induction (*Figure 1D*). Additionally, about 6.05% (n = 280) of egg chambers displayed fused egg chambers at early oogenesis, while degenerated egg chambers were observed in about half of all egg chambers

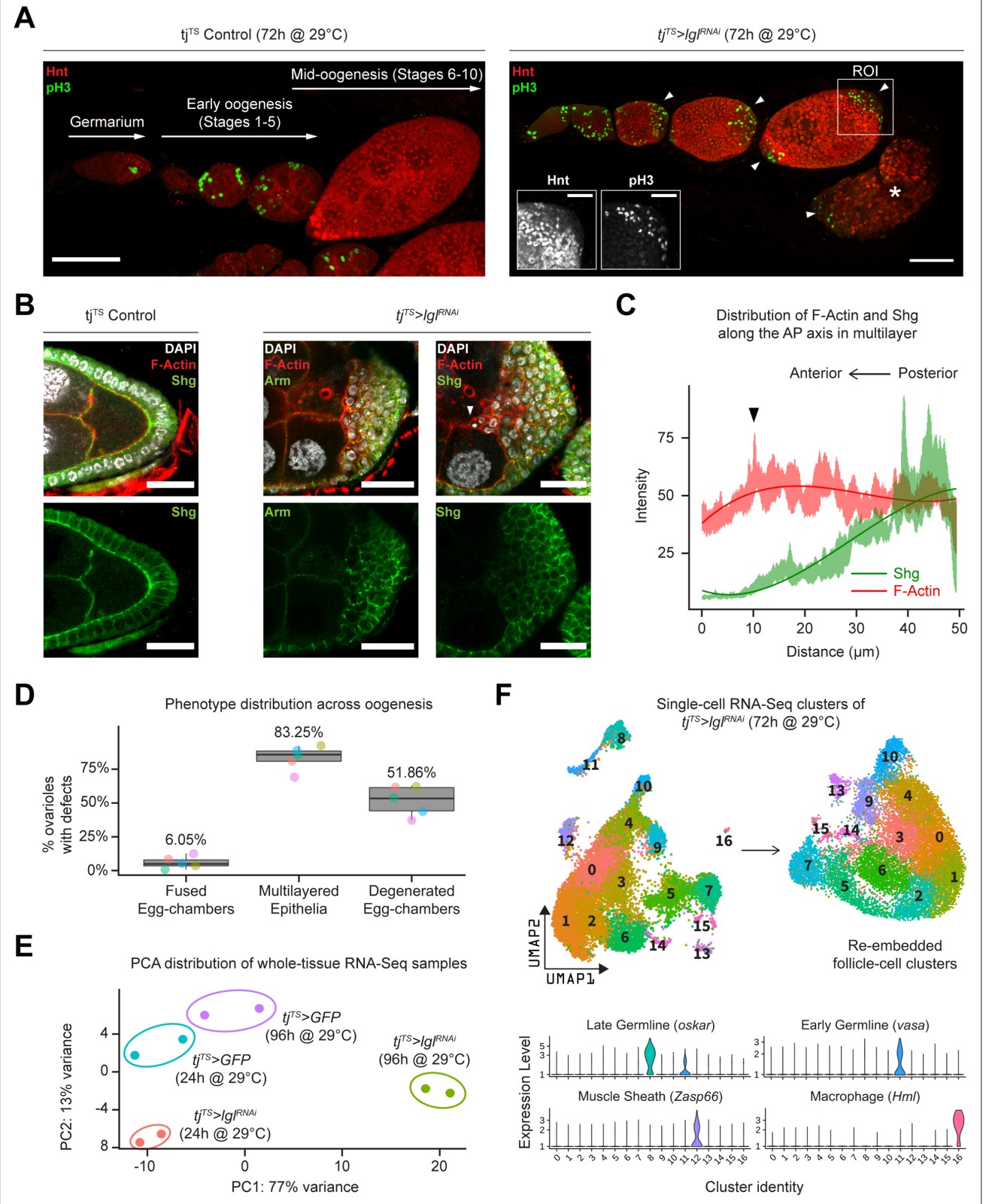

**Figure 1.** Inducing Lgl knockdown in follicle cells causes distinct phenotypic and transcriptomic changes. (**A**) Left: orthogonal projection of a single ovariole displaying individual egg chambers containing experimental control follicle cells at early and midoogenesis. Follicle cells at mitotic stages are infrequently detected by pH3 staining (green), while endocycling follicle cells at midoogenesis are labeled by Hnt staining (red). Right: ovariole containing egg chambers with Lgl-KD in follicle cells exhibit continued cell division (marked by pH3 staining in green) in cells that accumulate at egg

*Figure 1 continued on next page*

*Figure 1 continued*

chamber termini (arrowheads) at early-to-midoogenesis developmental transition and midoogenesis. Degenerated egg chambers containing dying germline cells are marked by asterisks (*). Scale bars: 50 μm. Distinct Hnt and pH3 staining within the Lgl-KD multilayers are highlighted for the region of interest (ROI) within the image. ROI scale bar: 20 μm. (**B**) Left: cross-section of the posterior egg chamber epithelia containing experimental control follicle cells that exhibit intact Shg (DE-Cad) staining at cell junctions. Middle and right: posterior multilayers of egg chambers containing Lgl-KD in follicle cells show declining enrichment (green) of Armadillo (Arm; middle panel) and Shg (DE-Cad; right panel) along the anterior–posterior (AP) axis (right to left). F-actin (red) is found enriched in cells at the apical-most layers; leading edge of the invasive front is indicated by an arrowhead. Nucleus is marked by DAPI (white). Scale bars: 20 μm. (**C**) Relative enrichment of F-actin (red) and Shg (DE-Cad; green) along the AP axis across the biggest distance between the apical-most and basal-most cells in the multilayer shown in (**B**) (right panel). Intensities are measured across a 10 μm thickness (Z-axis) and a trendline (Gaussian fit) is shown. The black arrowhead marks the leading edge corresponding to that shown in (**B**). (**D**) Box-and-whisker plot showing quantification of the different *tj$^{TS}$>lgl$^{RNAi}$* (72 hr in permissive temperature) phenotypes. Data was collected from five replicate trials (color-coded individually), consisting of 1250 intact ovarioles from a total of 165 flies. (**E**) Principal component analysis (PCA) plot showing the distribution of whole-tissue RNA-seq samples for *tj$^{TS}$* experimental controls kept for 24 hr and 96 hr in permissive temperature and their experimental counterparts containing *tj$^{TS}$>lgl$^{RNAi}$*. Each uniquely colored sample has two replicates that are grouped. (**F**) Overview of the single-cell (sc) RNA-seq workflow to isolate follicle cell-specific clusters from 14,537 *tj$^{TS}$>lgl$^{RNAi}$* ovarian cells, embedded on lower UMAP dimensions (top). Clusters containing nonepithelial cell types are identified by the enrichment of specific markers (bottom).

The online version of this article includes the following source data and figure supplement(s) for figure 1:

**Source data 1.** Raw data for (unnormalized) IntDen values of RFP, GFP, and DAPI enrichment.

**Source data 2.** Quantification of the stage-specific phenotypic characterization of ovarioles containing egg chambers with 96 hr Lgl-KD in their follicle cells.

**Figure supplement 1.** Lgl loss of function in follicle cells causes invasive multilayering and cell fate heterogeneity.

**Figure supplement 2.** Whole-tissue RNA-seq of samples containing multilayered Lgl-KD follicle cells.

**Figure supplement 2—source data 1.** Differentially expressed genes in 96h-Lgl-KD vs. others comparison.

**Figure supplement 3.** Single-cell RNA-seq of ovaries with 72h-Lgl-KD follicle cells.

**Figure supplement 3—source data 1.** Cluster-specific markers of the *tj$^{TS}$>lgl$^{RNAi}$* (72 hr) single-cell RNA-seq (scRNA-seq) dataset.

(51.86%; n = 280) at developmental stages beyond stage 9 or 10 during late oogenesis as a consequence of germline cell death (*Figure 1D*). Overall, these quantifications described the diverse egg chamber phenotypes that manifest upon Lgl-KD in follicle cells.

We next wondered whether the diverse ovarian phenotypes caused by Lgl-KD follicle cells were supported by significant positive gene-expression changes. To test this, we performed whole-tissue RNA sequencing (RNA-seq) analysis of ovarian tissues from both experimental control and Lgl-KD flies with RNAi induction for 24 hr (shorter) and 96 hr (longer periods of time). While 24h-Lgl-KD did not induce observable phenotypic changes, 96h-Lgl-KD displayed significant epithelial multilayering and germline cell death (*Figure 1—figure supplement 2A*). Principal component analysis (PCA) comparing all four samples separated the 96h-Lgl-KD sample from others across principal component 1 (PC1; 77% variance), indicating that significant transcriptomic differences separate the sample with the differential phenotype from the rest (*Figure 1E*). When compared with all these 'no-phenotype' samples collectively, several genes associated with late-stage oogenesis (e.g., *Vm26Ac*, *Femcoat*, *Cp15*, *yellow-g*, etc.) showed reduced expression, while expression of genes involved in actin binding (Molecular Function GO: 0003779; p=2.843 × 10$^{-3}$), locomotion (Biological Process GO: 0040011; p=3.748 × 10$^{-11}$), and cell periphery (Cellular Component GO: 0071944; p=2.904 × 10$^{-19}$) were found elevated in the 96h-Lgl-KD sample (*Figure 1—figure supplement 2B*, *Figure 1—figure supplement 2—source data 1*). Additionally, apoptotic cell clearance genes, such as *croquemort* (*crq*) (0.66 log$_2$FC; p=0.00015) and *draper* (*drpr*) (0.642 log$_2$FC; p=0.00027), were also upregulated in the 96h-Lgl-KD samples, likely in response to increased germline cell death at later stages of oogenesis (*Etchegaray et al., 2012*; *Franc et al., 1996*; *Franc et al., 1999*). While these results mostly agree with the quantified phenotype, they do not distinguish gene expression by the distinct phenotypes or individual cell types. To identify transcriptional changes specific to follicle cells, single-cell RNA-seq (scRNA-seq) was performed subsequently on cells isolated from 72h-*tj$^{TS}$>lgl$^{RNAi}$* ovaries (see 'Methods).

## Integrated analysis identifies unique transcriptomic clusters associated with Lgl-KD

On the basis of gene expression similarities, 16,060 ovarian cells of the $tj^{TS}$>$lgl^{RNAi}$ dataset were grouped into 17 clusters (*Figure 1F*, *Figure 1—figure supplement 3A–C*). Follicle cells were isolated for more targeted analyses by positively identifying nonepithelial cell types using previously described markers (*Jevitt et al., 2020*) (*vasa+* and *oskar+* early and late germline cells, *Zasp66+* muscle sheath cells and *Hml+* macrophage cell population known as the hemocytes) and removing them from the dataset. The remaining 14,537 Lgl-KD follicle cells were then integrated with 17,875 $w^{1118}$ follicle cells (that were processed similarly to remove nonepithelial cells) and were collectively assembled into 20 clusters (*Figure 2A*). These clusters were annotated according to the differential enrichment of stage- and cell-type-specific marker expression (*Figure 2B*). Consistent with the phenotypic observation that egg chambers degenerate post-midoogenesis around stage 8, clusters of cells belonging to egg chambers at stage 9 and beyond (clusters 9, 10, 11, 14, and 19) had lower proportions of Lgl-KD cells (*Figure 2C and D*). Significantly, we also identified clusters (clusters 7, 8, 13, 16, and 17) that were unique to the Lgl-KD dataset (*Figure 2E*). Our integration-based approach was thus able to separate clusters unique to the individual datasets, possibly representative of cells exhibiting the diverse stage-specific Lgl-KD phenotypes and the loss of cells from expected stages comparable to the $w^{1118}$ developmental lineage.

Keeping cluster identities intact, the $w^{1118}$ cells were removed and the 14,537 Lgl-KD follicle cells were re-embedded in the same UMAP space that was built using anchor-restricted principal components (PCs) upon integration (*Figure 3A*, left). The underlying lineage transitions among the cells of unique Lgl-KD clusters were then estimated by dynamically modeling their inherent RNA velocity (*Bergen et al., 2020*). The resultant stream of velocity vectors placed clusters 7, 13, 16, and 17 along a linear transition that terminated into cluster 8, while cluster 8 itself showed mixed lineage as was suggested by the nonuniform direction of velocity vectors (*Figure 3A*, right). The latent time experienced by the cells in these clusters as well as the likely terminal endpoints were then inferred from the estimated lineage, which subsequently suggested that clusters 13, 16 and the near-terminal cells of cluster 7 were the stable end points of the assumed lineage while cluster 7 displayed highly dynamic transcriptional transition (*Figure 3B*). Overall, given the uncompacted topology of certain cluster manifolds as well as the results from our linage analysis, we concluded that the unique clusters were a mix of stable terminal cell states as well as transitioning cell states (see 'Methods').

## Unique Lgl-KD clusters exhibit specific gene expression and regulon activity

To characterize clusters 7, 8, 13, 16, and 17 further, we identified the differentially expressed markers (*Figure 3C*). Apoptotic cell clearance markers *drpr* (*Etchegaray et al., 2012*) and *crq* (*Franc et al., 1999*), which were previously detected via bulk RNA-seq, were now specifically detected in the cells of cluster 8, while polar and border cell-associated markers (*Jevitt et al., 2020*), such as *unpaired* (*upd1*), *slow border cells* (*slbo*), and *singed* (*sn*), were found enriched in cluster 17 cells. We validated the expression of *sn* and *drpr* within the multilayers and found them to be expressed in the polar cells and in the follicle cells of egg chambers that associate with dying germline cells, respectively (*Figure 3D*). Border cell-specific expression of *sn* further implied that once the egg chambers have individualized, the polar to border cell fate is not disrupted upon Lgl-KD, despite the impairment of border cell migration. In contrast, clusters 7, 13, and 16 did not exhibit markers distinguishable from the normally developing cells from which they likely originate. While markers of immature, mitotic follicle cells (*Jevitt et al., 2020*) such as *Myc*, *Df31,* and *HmgD* were detected in cluster 13, it showed significant overlap with cluster 16, with the notable exception of the *Drosophila* Cdc25 homolog *string* (*stg*) expression. This difference in Stg expression led us to believe that clusters 13 and 16 represent immature (mitotic) and mature (endocycling) follicle cells. Additionally, very few specific markers were identified for cluster 7 as the markers either shared gene expression with (1) the main body follicle cells (expressing *Yp2* and *Yp3*), suggestive of a shared (and possibly, derivative) lineage, and (2) clusters 8, 13, and 16, exhibiting only quantitative differences between them (*Figure 3—figure supplement 1*). While such results are insightful, these observations somewhat highlight the limitations of marker validation to identify specific cells of the differential Lgl-KD phenotype.

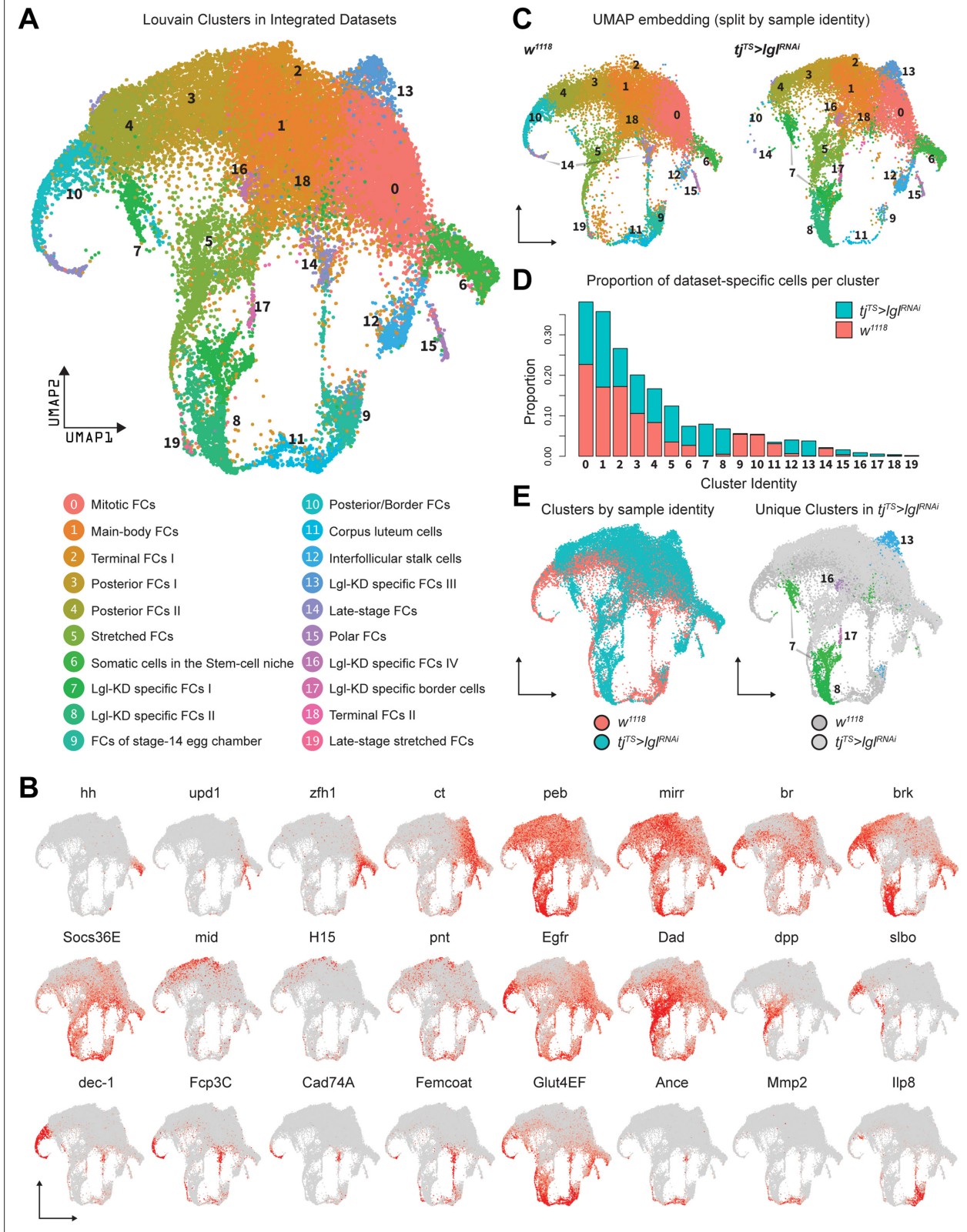

**Figure 2.** Integration of $w^{1118}$ and $tj^{TS}>lgl^{RNAi}$ single-cell datasets identifies sample-specific clusters. (**A**) UMAP plot of 17,874 $w^{1118}$ follicle cells integrated with 12,923 $tj^{TS}>lgl^{RNAi}$ follicle cells, grouped into 20 clusters, with their approximated identities listed below. (**B**) Canonical marker expression used to annotate the 20 integrated clusters reveals cluster-specific identities of the somatic cells of the stem cell niche (*hh*), polar follicle cells (*upd1*), stalk cells (*zfh1*), immature mitotic follicle cells (*ct*), mature postmitotic cells (*peb*), main body follicle cells (*mirr* and *br*), terminal follicle cells (*brk*, *Socs36E* and

*Figure 2 continued on next page*

Figure 2 continued

Egfr), posterior follicle cells (*mid, H15,* and *pnt*), stretched cells (*Dad* and *dpp*), border cells (*slbo*), follicle cells of the vitellogenic (*dec-1*) and choriogenic stages (*Fcp3C, Cad74A, Femcoat,* and *Glut4EF*), and the cells of the terminally fated corpus luteum (*Ance, Mmp2,* and *Ilp8*). (**C**) Distribution of clusters on the UMAP embedding is shown split by sample ID. (**D**) Bar plot showing the relative proportion of cells in each cluster. Each bar is further divided by the dataset of origin (*w*[1118] is represented by salmon color and *tj*[TS]>*lgl*[RNAi] by teal). (**E**) UMAP plot showing the overlap between cells from each dataset (left) and clusters unique to the 72h-*tj*[TS]>*lgl*[RNAi] dataset (right).

Instead of focusing on individual gene expression, we assessed the transcription factor (TF) regulon activity in the unique Lgl-KD clusters using SCENIC (*Aibar et al., 2017*). We specifically focused on cluster 7 (1144 cells), cluster 8 (913 cells), and cluster 13 (538 cells), and removed clusters 16 and 17 from the regulon analysis due to their low cell numbers (128 and 79 cells, respectively). While clusters 7 and 13 showed major differences in the enriched regulons, we found only two regulons (Hr4 and crc_extended) specific to cluster 7 (*Figure 3E*). Hierarchically grouping the clusters based on scaled regulon activity found cluster 7 to be more similar to cluster 8 than cluster 13, as was also evident from the relatively lower values for the enrichment of several regulons. To further explore this relatedness between the clusters, we evaluated regulon specificity (*Suo et al., 2018*) for each cluster (*Figure 3F*, *Figure 3—source data 1*). The five most specifically enriched regulons of cluster 7 included the TFs crc (regulon specificity score [RSS] = 0.42), Xbp1 (0.42), Pdp1 (0.41), BEAF-32 (0.41), and CrebA (0.41) and were identified alongside the AP-1 TFs *Jra* (0.4) and *kay* (0.39) that have been reported to drive tumorigenic JNK signaling upon polarity loss (*Bunker et al., 2015*). Comparable specificity scores were detected for kay (0.39), Jra (0.37), Xbp1 (0.36) CrebA (0.35), crc (0.33), BEAF-32 (0.32), and Pdp1 (0.27) in cluster 8, while cluster 13 showed lower specific activity for all the detected regulons. Consolidating the results from marker validation, RNA velocity-based lineage inference and the relative specificity as well as activity of detected regulons, a dynamic gene expression profile was observed in cluster 7 cells. These cells are derived from both mitotic cells (as velocity vectors can be seen transitioning from cluster 13 to 7) as well as endocycling cells (given the expression of main body follicle cell markers) and represent a transcriptomic state preceding that of the cluster 8 (that represents cells at a later developmental timepoint), as the enriched regulons show similar specificity but elevated activity from cluster 7 to cluster 8.

## Cluster 7 represents transient cells with heterogenous gene expression

Among the regulons exhibiting elevated activity in cluster 7 than cluster 8, the JNK signaling-associated AP-1 TFs *Jra* and *kay* have been previously implicated in polarity loss-induced metastatic tumor formation in the wing discs (*Bunker et al., 2015*; *Igaki et al., 2006*; *Külshammer et al., 2015*; *Uhlirova and Bohmann, 2006*). We then sought to assess the presence of tumorigenic JNK signaling in our Lgl-KD follicle cell model using quantitative real-time (qRT)-PCR. We specifically assessed the transcript levels of genes such as Ets at 21C (Ets21C), TNFα-receptor Grindelwald (grnd), Jra, kay, and the downstream target matrix metalloproteinase-1 (Mmp1), which are associated with tumorigenic JNK signaling network (*Andersen et al., 2015*; *Toggweiler et al., 2016*; *Uhlirova and Bohmann, 2006*) in whole ovaries with Lgl-KD in all follicle cells. Compared to the *tj*[TS] experimental control ovaries, we detected significant upregulation of *grnd* (GAPDH-normalized expression relative to the experimental control: 2.84 ± 0.122 standard error; p=0.0002), *Jra* (1.62 ± 0.059; p=0.0086), and *Mmp1* (2.79 ± 0.129; p=0.0002) and noticeable upregulation of *Ets21C* (1.88 ± 0.35; p=0.0861) and *kay* (1.58 ± 0.188; p=0.0837) in ovaries with Lgl-KD follicle cells (*Figure 3G*). Mapping the expression of *Ets21C*, *grnd*, *Jra*, and *kay* to specific Lgl-KD follicle cell clusters, we found that while both clusters 7 and 8 exhibit *Jra*, *kay*, and *grnd* enrichment, *Ets21C* is specifically detected in cluster 7, the terminal states of which also overlapped with dynamic *Mmp1* expression (*Figure 3H*, *Figure 3—figure supplement 2*). Collectively, our results indicate that cluster 7 consists of heterogenous tumorigenic cells, which could be further divided into transcriptomically homogenous groups to identify markers that could help us detect the precise location of these cells.

To explore cluster 7 heterogeneity, we subdivided its 1144 cells even further and obtained five transcriptomically similar neighborhoods (*Figure 4A*). The underlying lineage among these subclusters was then inferred from the inherent RNA velocity specific to cells in cluster 7. From the stream of velocity vectors superimposed on the cells embedded on UMAP space, we deduced that clusters 7_0 (293 cells) and 7_2 (193 cells) were the 'roots' of the inferred lineage that terminated into cluster

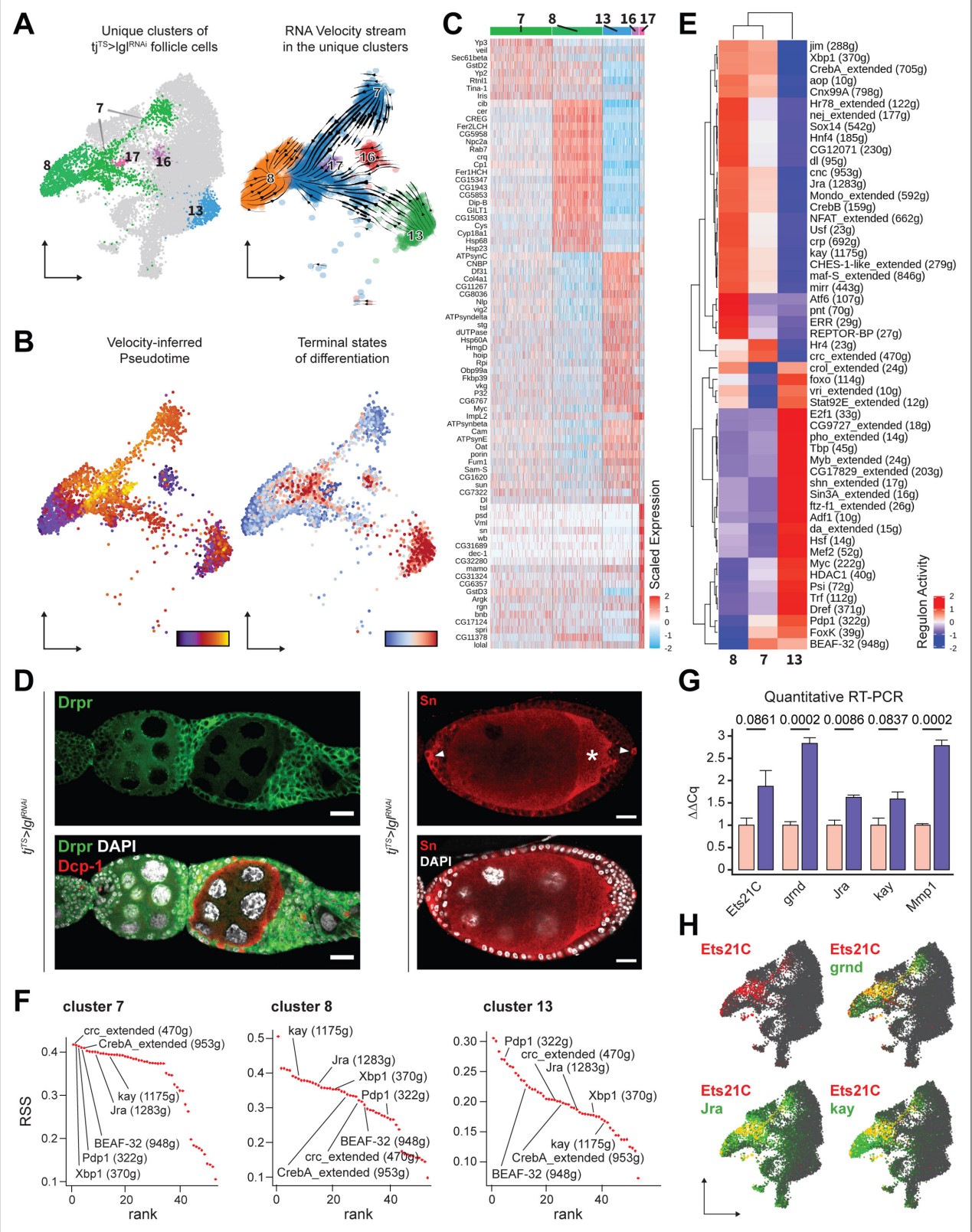

**Figure 3.** Unique clusters of *tj^TS>lgl^RNAi* dataset exhibit distinct gene expression and regulon activity. (**A**) Left: UMAP plot of the re-embedded *tj^TS>lgl^RNAi* follicle cells with the unique clusters being highlighted. Right: RNA velocity vectors superimposed on the embedded cells of unique *tj^TS>lgl^RNAi* clusters revealing the inferred lineage. (**B**) Left: cells are colored according to their arrangement on the velocity-inferred pseudotime from early (purple) to late (yellow). Right: cells are colored to denote terminal states of differentiation according to their position on the inferred lineage, where stable end points

*Figure 3 continued on next page*

*Figure 3 continued*

are colored in red and root cells are colored in blue. Observed noncompliance of inferred lineage and terminal states likely represents mixed population of cells. (**C**) Heatmap of the top 20 cluster-specific markers (if present) in the 72h-*tj^TS^>lgl^RNAi^* dataset. Selected genes are expressed in a minimum of 75% cells per cluster. Range of gene expression is scaled within +2 (red) to –2 (blue) log₂ fold change. (**D**) Confocal images showing Drpr (left) in green and Sn staining (right) in red. Sn+ polar cells are indicated by arrowheads, while oocyte is indicated by asterisks (*). Dcp-1 (red) is also shown in left panel to identify dying germline cells within degenerating egg chambers. Nucleus is marked by DAPI (white). Scale bars: 20 μm. (**E**) Heatmap of the scaled (and centered) activity scores of regulons in clusters 7, 8, and 13. Both columns (clusters) and rows (regulons) are clustered hierarchically and relative similarity between cluster 7 and cluster 8 regulon activity is inferred. (**F**) Regulon specificity score (RSS) rank plots show the specificity of regulon activity for clusters 7, 8, and 13. (**G**) Bar plot shows the relative mRNA levels of JNK signaling pathway components Ets21C, grnd, Jra, kay, and Mmp1 in *tj^TS^* control (N = 12 pairs of ovaries) and *tj^TS^>lgl^RNAi^* sample (N = 15) using quantitative RT-PCR. Bars representing the *tj^TS^* experimental control are colored salmon while *tj^TS^>lgl^RNAi^* is colored purple. Error bars represent Standard Error (SE) and the p-values obtained for individual t-test comparisons between samples are listed above each pairing. (**H**) Gene enrichment of *Ets21C* (red), *Jra*, *kay*, and *grnd* (green) is shown on the UMAP-embedded cells. Overlapping expression is colored in yellow.

The online version of this article includes the following source data and figure supplement(s) for figure 3:

**Source data 1.** Regulon specificity scores (RSS) of regulons enriched in clusters 7, 8, and 13 in *tj^TS^>lgl^RNAi^* (72 hr) single-cell RNA-seq (scRNA-seq) dataset.

**Figure supplement 1.** Validation of overlapping markers in Lgl-KD cells.

**Figure supplement 2.** Expression of JNK signaling components in 72h-Lgl-KD cells.

7_1 (291 cells), while clusters 7_3 (191 cells) and 7_4 (176 cells) were the intermediate states of transcriptional transition. To describe the regulatory relationships of TFs in all cluster 7 cells, we evaluated which of them might cooperate with each other by assessing the Connection Specificity Index (CSI) metric (**Fuxman Bass et al., 2013**). Hierarchically clustering the 16 regulons active in cluster 7 (including those that are suffixed '_extended' since they incorporate low-confidence TF associations and are inclusive of more genes), we inferred that the regulons could be organized into two larger clusters (**Figure 4B**). One of these highly correlated clusters contained both the AP-1 TF heterodimers *Jra* and *kay* along with the basic helix-loop-helix (bHLH) TF *Usf, ftz transcription factor 1* (*ftz-f1*) and the GATA-binding TF *serpent* (*srp*). The other cluster included the AP-1 interacting gene *activating transcription factor 3* (*Atf3*), nuclear receptor TFs *hormone receptor 4* (*Hr4*), *hepatocyte nuclear factor 4* (*Hnf4*), the bHLH transcriptional repressor *hairy* (*h*), *cyclic-AMP response element binding protein A* (*CrebA*), and *Ets21C*. Furthermore, when the relative activity of these regulons was compared between clusters, highly cooperative regulons showed similar extent of enrichment for each cluster (**Figure 4C**). For example, regulons associated with TFs Jra and kay showed low activity in clusters representing earlier timepoints, intermediate enrichment in the transitioning clusters and high activity in the terminal cluster. Similarly, the Ets21C regulon exhibit a pattern that was opposite to that shown by the AP-1 TFs. Overall, our comprehensive analytical approach not only identified processes specific to individual cells but also resolved regulon activity over time, as was experienced by the transitioning cluster 7 cells.

Next, we identified the differentially enriched markers of the different subclusters of cluster 7 to further characterize its underlying heterogeneity. Detecting specific markers, we annotated cluster 7_0 as mature, postmitotic follicle cells (*Yp1-3*+ and *Vm26Ab*+) and cluster 7_2 as immature, mitotic follicle cells (*stg*+, *Myc*+, and *HmgD*+), while cluster 7_1 expressed markers involved in apoptotic cell clearance (*drpr*+ and *crq*+) representing the degenerated egg chambers (**Figure 4D**). This conclusion was in agreement with the assignment of clusters 7_0 and 7_2 as 'roots' and 7_1 as the 'terminal' state, and we hypothesized that these cells were those that comprised the heterogenous multilayer. We were particularly interested in identifying the differentially enriched markers of cluster 7_3, which were composed of cells that transitioned from the mature cell cluster 7_0, as the underlying gene expression in them would likely represent the multilayered cells at the leading edge, given that the invasive front is composed of delaminating postmitotic follicle cells. We found several relatively enriched markers of cluster 7_3 that associated with the actomyosin cytoskeleton (*Act5C*, *Act42A*, *Vinc*, *capt*, *cib* and *sqh* and *Mlc-c*), and were part of the glutathione metabolic process (*GstD1*, *GstD2*, and *GstD3*), Rab protein signal transduction (*Rab1*, *Rab7*, and *RabX*), as well as the lysosomal trafficking pathway (*Rab7*, *cathD*, *Cp1*, *Cln3*, and *Swim*) (**Figure 4—source data 2**). Genes coding for the cytosolic glutathione S-transferase (Gst) family proteins are expressed under the control of the TF *cap-n-collar* or *cnc* (*Drosophila* Nrf2 homolog) (**Sykiotis and Bohmann, 2008**), which was also present

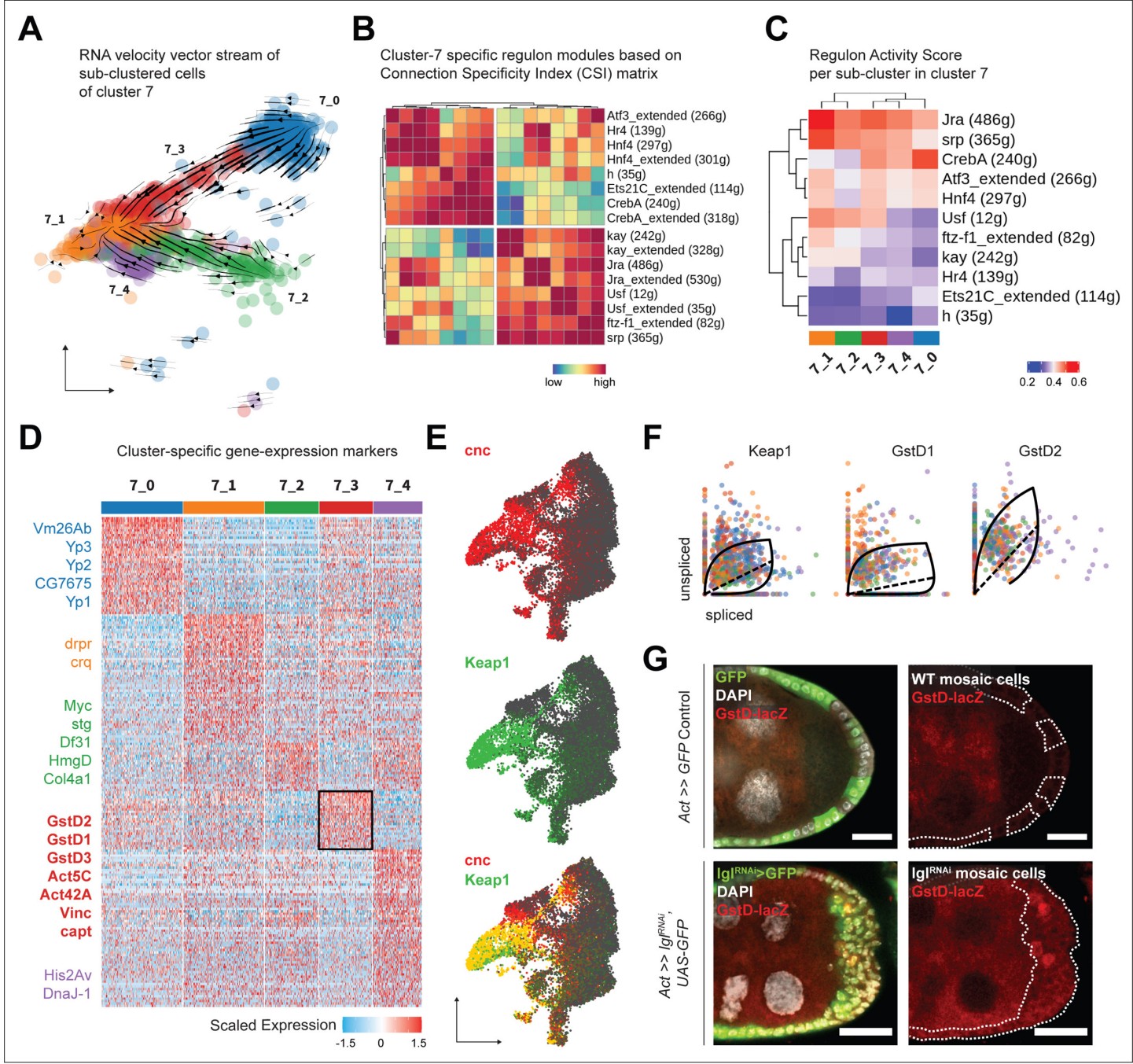

**Figure 4.** Cluster 7 cells exhibit heterogenous gene expression and regulon activity. (**A**) UMAP plot of subdivided cluster 7 cells, superimposed with RNA velocity vectors. (**B**) Heatmap representing regulon–regulon correlation based on Connection Specificity Index (CSI) of active regulon modules. Both high-confidence and low-confidence (marked by the '_extended' suffix) transcription factor (TF) associations are plotted. (**C**) Heatmap of the unscaled activity scores of regulons in the subclusters of cluster 7. (**D**) Heatmap of the (relatively) differentially expressed markers of each subcluster of cluster 7 cells, scaled within +1.5 (red) and –1.5 (blue) log$_2$ fold change. Select markers are mentioned, while those of cluster 7_3 (denoted by black border on the heatmap) are highlighted in bold. (**E**) Gene enrichment plot for *cnc* (red), *Keap1* (green), and their overlap (bottom). (**F**) Phase portraits showing dynamic behavior of genes in cluster 7 cells (colored by their subcluster ID as shown in panel **A**). Solid line represents the learned splicing dynamics while the dotted line represents the inferred gene expression steady state. *Keap1*, *GstD1*, and *GstD2* exhibit an acute increase in transcription in cluster 7 cells. (**G**) Reporter expression of *GstD-lacZ* is detected by β-gal expression (red) within the multilayered cells of *lgl^RNAi* follicle cells (green; clonal boundaries are marked by the dotted white lines). Nuclei is marked by DAPI (white). Scale bars: 20 μm.

The online version of this article includes the following source data and figure supplement(s) for figure 4:

**Source data 1.** Regulon specificity scores (RSS) of regulons enriched in the subclusters of cluster 7 in *tj^TS>lgl^RNAi* (72 hr) single-cell RNA-seq (scRNA-seq)

*Figure 4 continued on next page*

*Figure 4 continued*

dataset.

**Source data 2.** Differentially expressed markers of the subclusters of cluster 7 cells in *tj^{TS}>lgl^{RNAi}* (72 hr) single-cell RNA-seq (scRNA-seq) dataset.

**Figure supplement 1.** GstD-lacZ expression in Lgl-KD+Keap1-OE follicle cells and *w^{1118}* experimental control.

within the regulons of TFs Jra, kay, Usf, as well as Atf3 (*Source data 1*). From our results, we concluded that cluster 7_3 likely represented cells with elevated activation of the *cnc*-driven, stress-responsive Keap1-Nrf2 signaling pathway (*Yamamoto et al., 2018*).

We detected an enrichment of *cnc* and its endogenous inhibitor *Keap1* in clusters 7 and 8 of the Lgl-KD dataset (*Figure 4E*). In support of the transient expression inferred from detecting the elevated expression of Keap1-Nrf2 pathway target genes in cluster 7_3, RNA velocity analysis detected the dynamic behavior of *Keap1* as well as several Gst genes (*Figure 4F*). Using the *GstD-lacZ* enhancer trap reporter assay to validate Keap1-Nrf2 signaling activation (*Sykiotis and Bohmann, 2008*), we detected infrequent *β-gal* activity in subsets of cells within the multilayer (*Figure 4G*). We were therefore able to conclude that cluster 7 indeed represented the heterogenous cells of the multilayer, where the dynamic activation of Keap1-Nrf2 signaling was detected in a smaller subset of those cells.

## Keap1-Nrf2 signaling regulates invasive multilayering independent of its role in oxidative stress response

To determine how Keap1-Nrf2 signaling affects epithelial multilayering, we knocked down the expression of its upstream components *cnc* and *Keap1* in Lgl-KD cells (*Figure 5A and B*). In ovaries with Lgl-KD follicle cells, 86.35% (N = 793) ovarioles contained egg chambers with more than two layers of cells at mid-oogenesis, whereas Lgl-KD+Cnc-KD ovaries exhibited comparable multilayering in only 27.2% ovarioles (N = 964). Keap1-KD in Lgl-KD follicle cells exhibited a partial rescue of the Lgl-KD phenotype as only 35.46% ovarioles (N = 897) contained egg chambers with comparable multilayering. In both conditions, the timing of germline cell death was delayed resulting in development to proceed and form intact stage 10 egg chambers. These egg chambers continued to exhibit border cell migration defects, indicating that knocking down Cnc or Keap1 did not rescue the cell-autonomous defects of Lgl-KD cells and instead were negatively regulating some aspect of multilayer growth itself (*Figure 5A*, arrowheads). This reduction in Lgl-KD multilayering was also not caused by increased cytotoxicity induced by elevated oxidative stress following Keap1-Nrf2 signaling disruption as apoptosis was rarely observed in these cells (Dcp-1 staining not shown), thereby necessitating an alternate explanation for the rescue.

Contrary to our expectation that knocking down Keap1 (negative-regulator of Cnc) would likely promote Cnc-mediated transcription (*Itoh et al., 1999*), which would result in contrasting phenotypes to Cnc loss of function, genetic epistasis experiments involving *cnc* and *Keap1* knockdowns instead resulted in comparable phenotypes. To test how *Keap1* manipulation regulates Keap1-Nrf2 signaling pathway in the ovaries with and without Lgl-KD follicle cells, we assessed the relative levels of pathway components in the different genotypes using qRT-PCR (*Figure 5C*). We found that in the absence of Lgl-KD either overexpression or knockdown of *Keap1* did not significantly affect the levels of *cnc* but caused a decrease in the levels of GstD2, the bona fide target of Keap1-Nrf2 pathway. Manipulating Keap1 also did not cause a decrease in the number of egg chambers (data not shown) with migrating border cells that specifically express the GstD-lacZ reporter independent of Lgl-KD (*Figure 4—figure supplement 1*), thereby suggesting that the change is not because of the relative loss of sample. In the presence of Lgl-KD, however, overexpressing Keap1 caused an increase in GstD2 levels that was comparable to that observed in Lgl-KD alone. In these egg chambers with Lgl-KD+Keap1-OE follicle cells, we noticed a remarkable worsening of the Lgl-KD phenotype as the egg chambers appeared both multilayered as well as increasingly fused (*Figure 5D*). We also observed epithelial 'bridging' in several egg chambers (n = 119/185) at midoogenesis, where the multilayered cells formed invasive inroads through the recesses between germline cells that connected to the lateral epithelia, forming a bridge between two separate sections of the follicular epithelia (*Figure 5D*, arrowheads). While ectopic expression of UAS-CncC (longest isoform of wildtype Cnc; *Sykiotis and Bohmann, 2008*) also resulted in some enhancement of the Lgl-KD multilayering as well as fused egg chambers at early oogenesis (N = 39/121), it did not exhibit the 'bridging' phenotype in egg chambers at midoogenesis

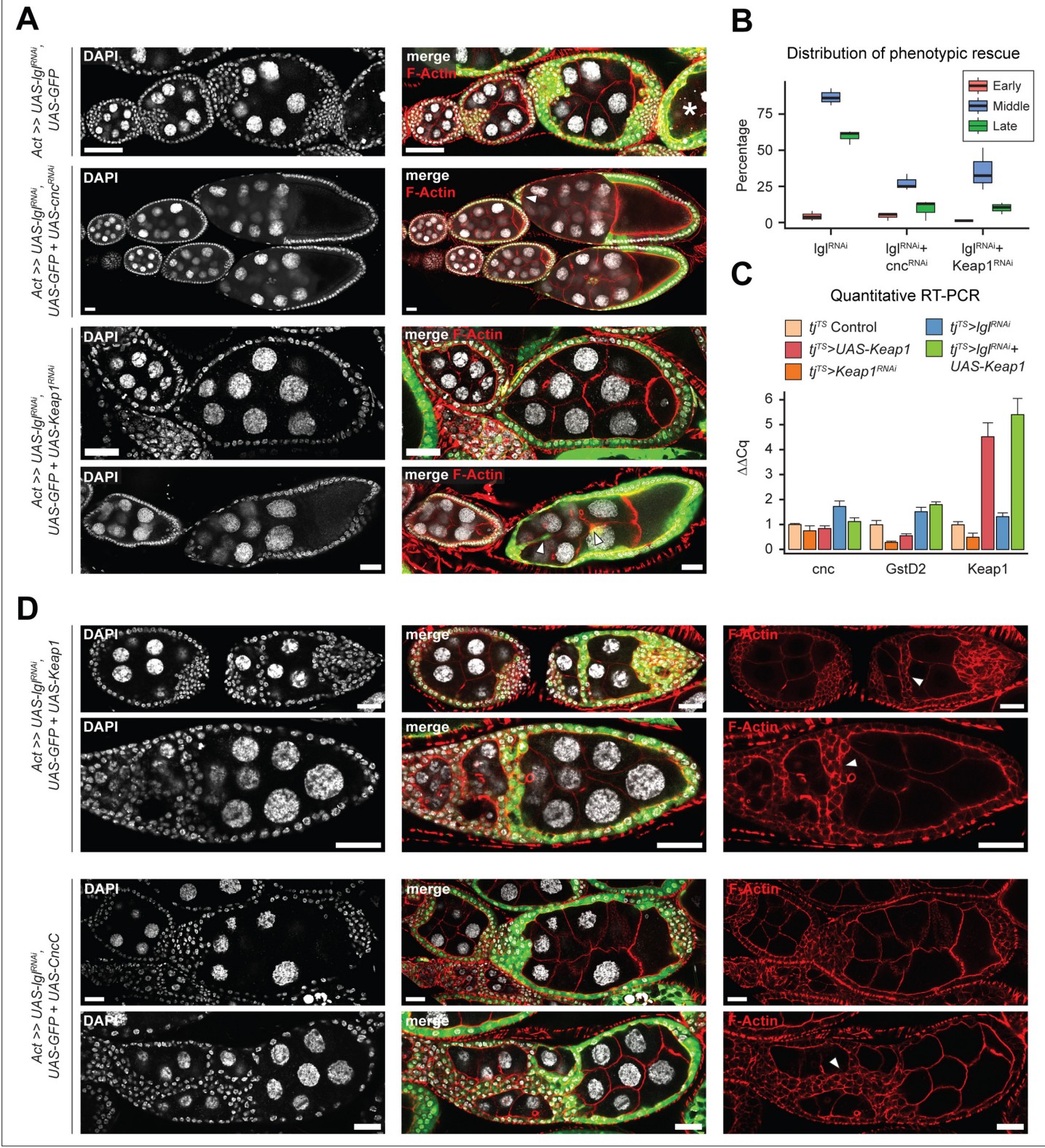

**Figure 5.** Keap1-Nrf2 signaling drives invasiveness of Lgl-KD multilayers. (**A**) Representative confocal images of ovarioles containing egg chambers with transgene expression in follicle cells (green) driving Lgl-KD (top), Lgl-KD+Cnc-KD (middle), and Lgl-KD+Keap1-KD (bottom panels). Degenerating egg chambers are marked by asterisks (*). Border cell migration defects are indicated by arrowheads. Nucleus is marked by DAPI (white). F-actin is marked by Phalloidin (red). Scale bars: 20 μm. (**B**) Box-and-whisker plot showing the quantification of phenotypic rescue at early (red), mid (blue), and late (green) oogenesis. For the multilayering phenotype at midoogenesis, we have only counted instances of >2 follicular layers. (**C**) Bar plot showing

*Figure 5 continued on next page*

*Figure 5 continued*

relative expression levels of genes involved in the Keap1-Nrf2 signaling pathway in relevant genotypes (N = minimum 10 pairs of ovaries). Samples are color-coded as shown in legend and the error bars represent Standard Error (SE). (**D**) Confocal images of ovarioles with Lgl-KD+Keap1-OE (above) and Lgl-KD+CncC-OE (below) in follicle cells (green). Arrowheads mark the epithelial bridging or fusion phenotype. Nucleus is marked by DAPI (white). F-actin is marked by Phalloidin (red). Scale bars, 20 μm.

The online version of this article includes the following source data and figure supplement(s) for figure 5:

**Source data 1.** Quantification of the rescue of stage-specific Lgl-KD phenotypes upon Cnc-KD and Keap1-KD.

**Figure supplement 1.** Epithelial multilayering and invasiveness are increased in Lgl-KD+CncC-OE egg chambers.

(*Figure 5D*). However, that the fused egg chamber phenotype occurred due to increased invasiveness, and not as a consequence of two individualized egg chambers fusing at the ends due to improper stalk cell differentiation, was evident from the observation that nurse cell nuclei were abnormally compressed by the invasive fronts of collectively migrating Lgl-KD+CncC-OE follicle cells (*Figure 5—figure supplement 1A*). Phenotypic enhancement in Lgl-KD+CncC-OE was also verified by observing increased multilayer formation in egg chambers at midoogenesis in 22.85% intact ovarioles (N = 16/70) at 2 days after heat shock (AHS) induction, which is an insufficient amount of time for Lgl-KD to exhibit multilayering (only 2% egg chambers showed multilayering; N = 50 ovarioles) (*Figure 5—figure supplement 1B*). We believe that the CncC-mediated enhancement is partially caused by the autonomous activation of Keap1 in the UAS-CncC-expressing cells (*Sykiotis and Bohmann, 2008*) and that the increased egg chamber fusion is a phenotype specific to the earlier developmental stages. Nonetheless, given that the invasive phenotype was not regulated by antagonizing functions of Keap1 and Cnc, the increased invasiveness of Lgl-KD by Keap1 overexpression was likely controlled by the noncanonical functions of Keap1-Nrf2 signaling outside its role in oxidative stress response.

## Elevated Keap1 increases collective invasiveness of multilayered cells

Unlike the Lgl-KD multilayers, in the egg chambers with Lgl-KD+Keap1-OE or Lgl-KD+CncC-OE enhanced invasion, we noticed gaps within the multilayers, indicating a probable loss of cell–cell adhesion (*Figure 6A*). We therefore sought to describe the cell junctions in these multilayers using antibodies against junctional proteins Shg (E-Cad) and Arm (β-Cat). In Lgl-KD+Keap1-OE egg chambers, a relative decrease in Shg and Arm enrichment was detected at the junctions of apically invading multilayered cells when compared to that in the basal-most layer (*Figure 6A*). However, the delaminated Lgl-KD+Keap1-OE cells did not exhibit a complete loss of cell–cell adhesion as Shg and Arm staining continued to be present at cell–cell junctions. This observation implied that these cells maintained cellular junctions, thereby explaining their collective mode of invasion and requiring alternate explanations for the occurrence of gaps.

We measured the relative proportion of delaminated epithelial volume in total volume of intact egg chambers containing Lgl-KD and Lgl-KD+Keap1-OE in follicle cells. We found that elevating *Keap1* in Lgl-KD background induced a significant (*p*=0.02) but variable increase (40.3%; N = 10 egg chambers at stage 8) in the delaminated epithelial volume compared to that of the Lgl-KD (21.8%; N = 9) alone (*Figure 6B*). The occurrence of gaps within the delaminated epithelia suggested that the multilayered cells possibly did not expand evenly as uniform growth would tend to fill any available space. Indeed, mitotic marker phospho-histone 3 (pH3) was also not detected within the invading cells of the multilayered epithelia (*Figure 6C*, middle panel), further suggesting that the increase in invasive epithelial volume was not caused primarily by overproliferating cells pushing out the delaminating epithelia. Instead, this phenotype was likely caused by an enhancement of invasive behavior.

We assessed whether the collectively moving cells displayed polarized invasion or multicellular streaming by observing F-actin distribution in the invasive cells (*Friedl et al., 2012*). While F-actin was generally found elevated in the delaminated epithelial bridges formed by Lgl-KD+Keap1-OE cells, when invasive multilayers displaying incomplete bridging were specifically observed, we noticed significantly increased F-actin staining at the leading edge formed by blunt multicellular tips (*Figure 6C*). Quantifying the relative F-actin intensity in the delaminated epithelia of both Lgl-KD (N = 13 invasive multilayers from 10 total egg chambers) and Lgl-KD+Keap1-OE (N = 18) egg chambers at midoogenesis revealed (1) an increase in F-actin enrichment in the multilayered epithelia compared to the monolayer and (2) a highly significant (p=0.00053) enhancement of F-actin intensity in the

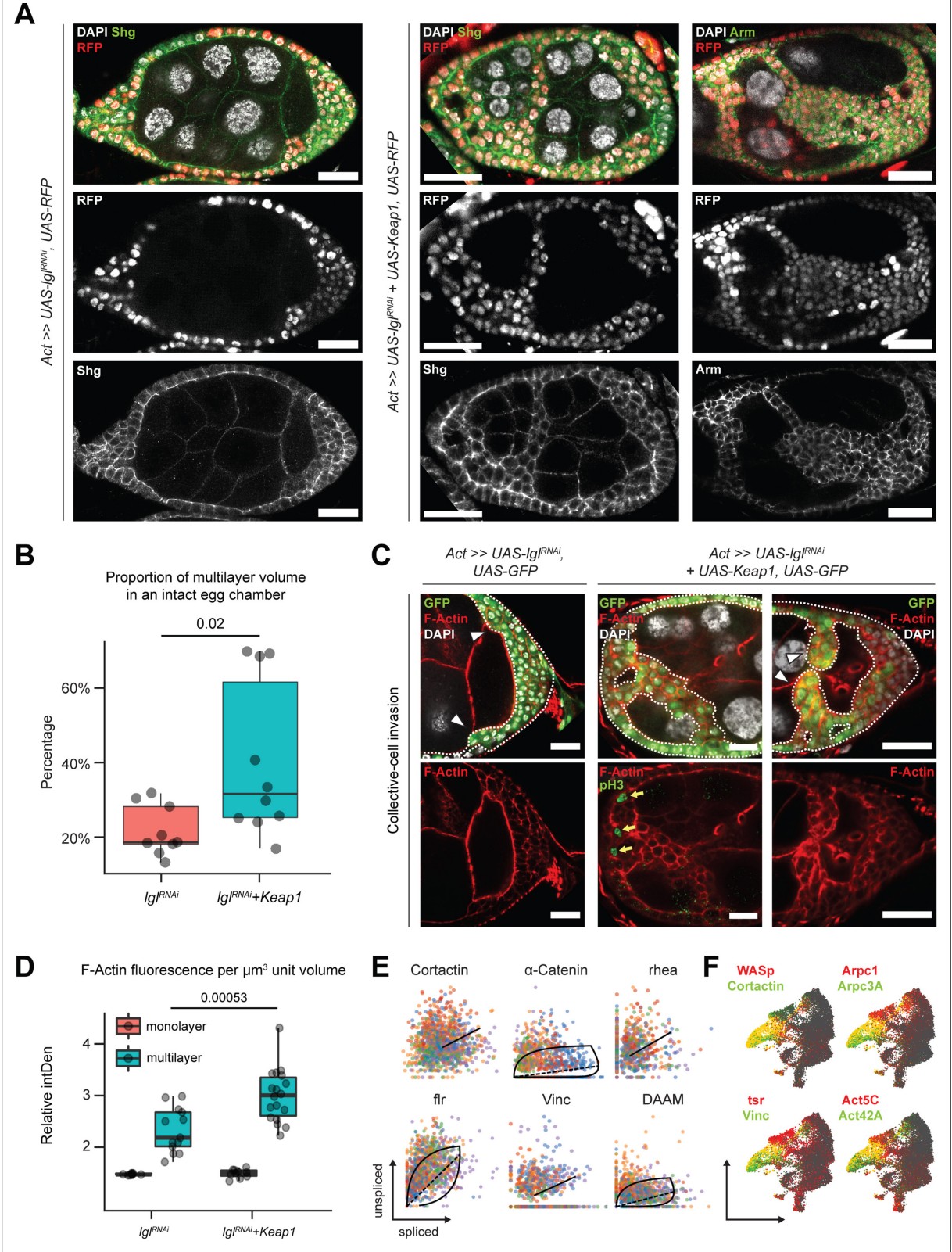

**Figure 6.** Invading Lgl-KD+Keap1-OE cells exhibit extensive cytoskeletal remodeling. (**A**) Confocal images of egg chambers with Lgl-KD (left) and Lgl-KD+Keap1-OE (middle and right panels) in the follicle cells (red; shown in the middle row). Egg chambers are stained with Shg (DE-Cad; left and center columns) and Arm (right column) (green). Nucleus is marked by DAPI (white). F-actin is marked by Phalloidin (red). Follicle cells with transgene expression are separately shown in the middle row, while Shg and Arm staining intensities are shown in the bottom row. Scale bars, 20 µm. (**B**) Box-and-

*Figure 6 continued on next page*

*Figure 6 continued*

whisker plot showing the proportion of delaminated epithelial volume of Lgl-KD (N = 9 egg chambers) and Lgl-KD+Keap1-OE (N = 10) follicle cells compared to the volume of the entire egg chamber. The p-value ($p$=0.02) obtained from t-test comparing the two samples is listed on top. (**C**) Confocal images of collectively invading follicle cells (green) in Lgl-KD (left) and Lgl-KD+Keap1-OE (middle and right). Nucleus is marked by DAPI (white). F-actin is marked by Phalloidin (red). Invasive fronts are marked by arrowheads. Positive pH3 staining (green in bottom middle panel) shows mitotically dividing cells, marked by yellow arrows. Scale bars, 20 μm. (**D**) Box-and-whisker plot to show the relative intensities of F-actin in the monolayer (red) and multilayers (teal) of egg chambers with Lgl-KD (N = 10 egg chambers) and Lgl-KD+Keap1-OE (N = 10) follicle cells. Individual egg chambers are shown as distinct dots. The p-value ($p$=0.00053) obtained from t-test comparing the two samples is listed on top. (**E**) Phase portraits of cytoskeletal genes in cluster 7. Straight line indicates that the gene exhibits steady-state dynamics, while curved solid line with dashed straight line represents upregulation. Cells are colored according to their subcluster identity. (**F**) Gene enrichment plot shows overlapping enrichment of actin-remodeling genes in cluster 7 cells.

The online version of this article includes the following source data and figure supplement(s) for figure 6:

**Source data 1.** Related to *Figure 6B*.

**Source data 2.** Raw data for the IntDen values of F-actin enrichment in the monolayered and multilayered follicle cells expressing Lgl-KD and Lgl-KD+Keap1-OE.

**Figure supplement 1.** Oxidative stress response is not detected in cluster 7 cells.

---

Lgl-KD+Keap1-OE multilayers compared with those in Lgl-KD (*Figure 6D*). The presence of a defined actin-rich leading edge with multiple cells at the tip indicated that the cells collectively undergo polarized collective invasion.

Collectively invading cells exhibit rapid turnover of cytoskeletal proteins, as well as invadipodial structures in the leading edge. From our RNA velocity analysis of the scRNA-seq dataset, dynamic behavior of several genes coding for proteins that mediate interactions between integrins and filamentous actin (e.g., *α-Catenin*, *Vinculin,* and *cofilin*) was detected (*Eddy et al., 2017*) along with those that regulate actin polymerization (*Krueger et al., 2019*; *Molnár et al., 2014*; *Poukkula et al., 2014*) (such as *Cortactin*, *flr,* and *DAAM*) in cluster 7 cells (*Figure 6E*, *Source data 2*). These genes have known functions in focal adhesion and are required for the formation and maintenance of invadopodia (*Aughey et al., 2016*; *Eddy et al., 2017*). We also detected enrichment of genes that support leading-edge protrusion (*Eddy et al., 2017*) such as *WASp*, *Cortactin*, *Vinculin*, *tsr* (*Drosophila* Cofilin), individual actin molecules (*Act5C* and *Act42A*), and specific genes of the Arp2/3 actin-nucleator complex (*Arpc3A* and *Arpc1*) (*Figure 6F*). Dynamic regulation of cytoskeletal genes in these cells further supports the assumption that delaminating Lgl-KD cells exhibit leading-edge invasion, and increasing Keap1 expression likely promotes this invasive behavior by extensive remodeling the actin cytoskeleton.

## Discussion

Our purpose in this study was to identify genes that regulate changes in cell plasticity upon apicobasal-polarity loss in epithelial cells, which is arguably one of the earliest characteristics of tumor cells (*Moreno-Bueno et al., 2008*; *Royer and Lu, 2011*). In this study, polarity loss in follicle cells was induced by knocking down the expression of basolateral polarity protein Lgl, which is also known as a neoplastic tumor suppressor gene since its deletion causes neoplastic transformation of epithelial tissues (*Froldi et al., 2008*). Orthologs of *l(2)gl* have been identified in both mice (*mgl-1*) (*Tomotsune et al., 1993*) and humans (*Hugl-1*) (*Strand et al., 1995*), and conservation of its tumor-suppressive role is supported by the observation that ectopic Hugl-1 expression could rescue *l(2)gl* mutant phenotype in *Drosophila* (*Grifoni et al., 2004*). Reduced expression of Hugl-1 has been detected in as many as 62% of samples in a comprehensive cohort of human solid tumors (*Grifoni et al., 2004*) and is associated with disease progression in cancers such as hepatocellular, colorectal, pancreatic, endometrial, as well as lung squamous cell carcinoma (*Biesterfeld et al., 2012*; *Lu et al., 2009*; *Matsuzaki et al., 2015*; *Schimanski et al., 2005*; *Tsuruga et al., 2007*). In metastatic melanoma, where Hugl-1 expression is also found decreased from that for normal skin, induction of Hugl-1 expression caused reduced cell migration and stress-induced cellular detachment, but did not induce proliferation (*Kuphal et al., 2005*). We believe that our observations of tumor-like behavior in the *Drosophila* follicle cell model support several of these conclusions from mammalian studies and our in-depth characterization of the

underlying transcriptome at single-cell resolution represents an important step toward developing a unified understanding of gene expression changes driving early tumorigenesis following polarity loss.

Among our primary findings is the identification of a heterogenous group of cells that transcriptomically deviate from their corresponding cells of origin. Given the absence of distinguishable markers (compared to the normally developing cells of origin), we hypothesized that these clusters represent divergent cell states and not unique cell types. Indeed, the cells in these clusters are characterized by only a relative enrichment of differentially expressing, tumorigenic stress signaling markers that have been previously identified from studies using the wing-disc tumor model (*Bunker et al., 2015*; *Hamaratoglu and Atkins, 2020*; *Mundorf et al., 2019*). Validating the expression of these markers using expression reporters and antibodies against specific proteins, we detected them either in the multilayered cells or in the polarity-deficient follicle cells at late stages of aborted development. Among the earliest distinguishable phenotypes, activation of Keap1-Nrf2 signaling pathway was detected within the multilayered cells, which was likely a response to the elevated ER stress (evidenced by an enrichment of *crc*, *Xbp1*, and several CREB/ATF-associated regulons in those cells) as has also been previously reported to happen (*Wortel et al., 2017*). This, however, does not occur on account of significant build-up of oxidative stress as neither the superoxide-detector dihydroethidium (DHE) nor the common oxidative stress response genes were detected in the multilayers or in cluster 7, respectively (*Figure 6—figure supplement 1*). Nonetheless, the role of oxidative stress cannot be ruled out entirely since a stressed ER also causes the accumulation of reactive oxygen species (ROS) (*Limia et al., 2019*; *Wortel et al., 2017*) that would directly impact Keap1 structure, allowing increased Nrf2-mediated transcription. Considering this redundancy in signaling regulation, it is difficult to ascertain the direct cause of Keap1-Nrf2 pathway activation in the multilayered cells from these experiments alone. Our conclusions from this study therefore warrant further investigations to understand the precise cause of Keap1-Nrf2 signaling activation in the cells with polarity loss.

Given that the role of Keap1-Nrf2 signaling in cancers is more commonly investigated by using loss-of-function mutants of *KEAP1* (*Wu and Papagiannakopoulos, 2020*), the role of its *wildtype* protein in regulating tumor cell behavior remains largely unexplored. Outside its role as a redox sensor, limited studies have indeed identified additional roles of Keap1 in regulating invasive cell behavior (*Rachakonda et al., 2010*) and maintaining cytoskeletal stability (*Kang et al., 2004*; *Yamaguchi and Condeelis, 2007*). *Wildtype* Keap1 has been shown to inhibit single-cell motility by strengthening stress fibers bundles via upregulated RhoA activity (*Wu et al., 2018*), but also stabilizing focal adhesion assemblies by reducing their turnover (*Dinkova-Kostova et al., 2005*; *Velichkova and Hasson, 2003*). Reduction in focal adhesion turnover maintains mature adhesions that are resistant to disassembly as cells change shape and protrude (*Fessenden et al., 2018*; *Oakes et al., 2012*). Indeed, the observation of Keap1-induced enhancement of Lgl-KD cell invasion having high F-actin enrichment likely reflects these autonomous cytoskeletal changes. Our study, therefore, attributes a highly tumor-relevant function to Keap1 as it stands to play during the earliest stages of locally growing tumors, during which the cellular burden of mutations is low (*Bozic et al., 2010*). Recently, dynamic regulation of Nrf2 within the collectively invading cancer cells has been shown to also have an effect on the metastable states of cancer cells displaying partial EMT (*Bocci et al., 2019*; *Vilchez Mercedes et al., 2022*; *Zhou et al., 2016*). It is therefore also possible that in our experiments Nrf2 expression regulates the metastability of Lgl-KD cells by indirectly modulating Keap1 as a result of which the invasive behavior of the delaminated tissue on the whole is likely being determined.

Conclusions from existing investigations of Keap1's role in cytoskeletal regulation are often confounded by the lack of a standardized approach to define semantics relevant to the distinct invasion/migration patterns in cells and how they may relate to the different steps of metastatic progression. Applying a framework developed for cancer cells to address potential mechanisms of migratory behavior (*Friedl et al., 2012*), we further characterized the invasive behavior of multilayered cells. Enriched F-actin at the leading edge of invasive fronts is a hallmark of invasive membrane protrusions of leader cells in a collectively migrating cell cohort (*Eddy et al., 2017*) and is frequently observed in tumor buds (*Grigore et al., 2016*). In our observations, we concluded that the delaminated cells showed leading-edge behavior based on the differential enrichment of F-actin in cells at the invasive front. Moreover, the presence of holes within the epithelia – which was not observed in Lgl-KD multilayers alone (*Goode et al., 2005*; *Szafranski and Goode, 2007*) – further suggested that the delaminating cells exhibit a differential capacity to migrate as the holes would likely have been filled if

every cell could either exert independent migratory force or exhibit uniform proliferation to compensate for the increased invasiveness. It is therefore plausible that the increase in polarized invasiveness is able to outrun the pace of cell division, leaving behind gaps in the delaminated epithelia. Despite strong evidence suggesting that ectopic Keap1 drives leading-edge-directed collective invasion of Lgl-KD multilayers, our conclusions are mildly tempered by the constraints of available space within the egg chambers, which limits the ability to separate collective invasion maintained by weak cell–cell adhesions and random movements of cells in the narrow passage between the germline cells. Time-lapse or live imaging of this invasive behavior could be used in subsequent studies to understand the underlying mechanism better. Nonetheless, our study presents an innovative analytical approach, which identifies novel regulators of tumor cell behavior in vivo in the polarity-deficient *Drosophila* follicular tumor model.

## Methods

### Fly husbandry

Flies were reared under standard lab conditions at 25°C and were fed dry yeast a day before dissections. For temperature-sensitive RNAi experiments, *tj-Gal4* driver was combined with *tub-Gal80^ts* and the resulting flies were reared at 18°C to repress spurious Gal4 activity and transferred to 29°C for transgene induction. These flies were kept at 29°C for the required number of days, following which they were dissected in phosphate-buffered saline (PBS). Combining *tj-Gal4, UAS-GFP, tub-Gal80^ts* with *UAS-l(2)gl^RNAi, UAS-Dcr2* resulted in increased genetic lethality and only a few flies were recovered. Therefore, *tj-Gal4* lines without a fluorescence reporter were used to generate a stable line and the resultant phenotype was tested for comparable results. For FLPout experiments, flies were reared at 25°C and were heat-shocked at 37°C for 20 min, fed with dry yeast 2 days AHS, and were dissected 3 days AHS.

### Fly stocks

1. BDSC stocks:
   UAS-Lgl-RNAi: y[1] v[1]; P{y[+t7.7] v[+t1.8]=TRiP.HMS01905}attP40 (#38989), UAS-Keap1: y[1] w[67c23]; P{y[+mDint2] w[+mC]=EPgy2}Keap1[EY02632] (#15427), UAS-Keap1-RNAi: y[1] sc[*] v[1] sev[21]; P{y[+t7.7] v[+t1.8]=TRiP.HMS02180}attP40 (#40932), UAS-cnc-RNAi: y[1] v[1]; P{y[+t7.7] v[+t1.8]=TRiP.JF02006}attP2 (#25984), hsFLP: P{ry[+t7.2]=hsFLP}1, y[1] w[1118]; Dr[Mio]/TM3, ry[*] Sb[1] (#7), Act>y>Gal4, UAS-GFP: y[1] w[*]; P{w[+mC]=AyGAL4}25P{w[+mC]=UAS-GFP.S65T}Myo31DF[T2] (#4411).
   hsFLP;; Act>CD2>Gal4, UAS-hRFP: w[*]; P{ry[+t7.2]=Act5C(FRT.polyA)lacZ.nls1}2, P{w[+mC]=Ubi-p63E(FRT.STOP)Stinger}9F6/CyO; P{w[+mC]=GAL4-Act5C(FRT.CD2).P}S, P{w[+mC]=UAS-His-RFP}3/TM3, Sb[1] (modified from #51308).
2. VDRC stocks:
   UAS-l(2)gl-RNAi (#51247).
3. Kyoto stocks:
   tj-Gal4: y[*] w[*]; P{w[+mW.hs]=GawB}NP1624/CyO, P{w[-]=UAS-lacZ.UW14}UW14 (#104055).
4. Others:
   GstD-lacZ and UASt-CncC (*Sykiotis and Bohmann, 2008*).

### Immunofluorescence staining and imaging

Flies were dissected at room temperature in 1× PBS, and the ovaries were fixed for 15 min in 4% paraformaldehyde (PFA). Fixed ovaries were then washed three times in 1× PBT (PBS with 0.2% Triton X-100) for 20 min per wash and were incubated for 1 hr with blocking solution (1× PBT with 0.5% BSA and normal goat serum). Incubation with primary antibody, diluted in blocking solution, was performed overnight at 4°C. The primary antibody-stained ovaries were again washed three times with 1× PBT the following day and were subsequently incubated with secondary antibodies diluted in blocking solution for 2 hr at room temperature. After washing again three times with 1× PBT, and once with PBS, the ovaries were dyed with DAPI (Invitrogen, 1 µg/mL) to stain the nuclei. Samples were finally mounted on microscopic slides after adding 80% glycerol mounting solution. The following antibodies or dyes are mentioned in this article.

1. Developmental Studies Hybridoma Bank (DSHB):
   Mouse anti-Arm (N27A1, 1:40 dilution), mouse anti-Cut (2B10, 1:30), rat anti-Shg (DCAD2, 1:20), mouse anti-Hnt (1G9, 1:15), rat anti-Mmp1 (1:1:1 mixture of 3B8, 3A6, and 5H7, 1:40), mouse anti-Sn (sn7C, 1:25), and mouse anti-Drpr (5D14, 1:50).
2. Promega:
   Mouse anti-β-gal (PAZ3783, 1:500).
3. Millipore:
   Rabbit anti-pH3 (06-570, 1:200).
4. Invitrogen:
   Phalloidin Flour 546 and 633 (A22283 and A22284; 1:50)
5. Secondary antibodies:
   Alexa Fluor 488, 546, and 633 (1:400, Molecular Probes).

For real-time DHE staining, $tj^{TS}>lgl^{RNAi}$ ovaries (n = 20) – with no UAS-tagged fluorescence reporter – were dissected in Grace's Insect Basal medium (VWR; #45000-476), and incubated in DHE (Invitrogen, D1168; diluted in 0.1 µmol/L DMSO) for 5 min. Following DHE incubation, ovaries were fixed and standard protocol for immunofluorescence were followed.

All images were acquired using the Zeiss LSM 800 confocal microscope and the proprietary Zeiss microscope software (ZEN Blue). Images comparing different samples under comparable experimental conditions were obtained using the fixed settings for image acquisition. Final images were processed and analyzed using the Fiji ImageJ (*Schindelin et al., 2012*) open-source software. Images were organized into figures using Adobe Illustrator software.

## Quantitative real-time (qRT)-PCR

Entire ovaries were lysed in TRIzol (Invitrogen) and the RNAs were prepared according to the manufacturer's protocol and quantified using NanoDrop. For each sample, 1 µg of mRNA was reverse-transcribed using oligo-dT-VN primers and ImProm-II as the reverse transcriptase (Promega) in triplicate. Real-time quantitative amplification of RNA was performed using the Sybr Green qPCR Super Mix (Invitrogen) and the iQ5 Real-Time PCR Detection System from Bio-Rad according to the manufacturer's protocol. The relative expression of indicated transcripts was quantified with the CFX_Manager Software (Bio-Rad) using the 2[−ΔΔC(T)] method. According to this method, the C(T) values for the expression of each transcript in each sample were normalized to the C(T) values of the control mRNA (*GAPDH*) in the same sample. The values of untreated cell samples were then set to 100%, and the percentage transcript expression was calculated.

The following primer pairs have been used in this article:

Ets21C-F: GAGGCCGATTAATGCCATGC
Ets21C-R: AGTTGAGGGCGGGTAATTGG
grnd-F: TCGGTCAGGAAGTTGAGTGC
grnd-R: CGCACAGAAACGCATCGTAG
Jra-F: AACACATCCACCCCGAATCC
Jra-R: CCTTGGTGGGGAACACCTTT
kay-F: TTTCTGCCCGCCGATCTAAG
kay-R: GTTGCCGAGGATAAGATTGCG
Mmp1-F: CAAGTTGGACGAGGACGACA
Mmp1-R: GTAGGCCTCAGCTGGTTTGT
cnc-F: CCACAACACCACCGGGAATA
cnc-R: ATGTGGCGTGAGGAAAGTGT
GstD2-F: CCGGATCGGATGAGGACTTG
GstD2-R: TTCGAACGTGGAGACAGTGG
Keap1-F: TTCCTGCAGCTTTCGGCATA
Keap1-R: GCTCCTCCTGCACATTCAGT

## Whole-tissue RNA-sequencing sample preparation for sequencing

Flies of the genotypes *tj>GAL4^TS^, UAS-GFP* (experimental control) and *tj>TSGAL4^TS^, UAS-l(2)gl^RNAi* (no GFP) (experiment) were raised in 18°C for 2–3 days post-eclosion. They were then transferred to 29°C for either 24 hr (1 day) or 96 hr (4 days) to induce the multilayering of follicle cells for short and long

periods of time. All female flies had access to males in their vials and were given yeast supplement 24 hr prior to dissection to facilitate oogenesis. Ovaries were dissected from 40 flies in complete medium Grace's Insect Basal medium (VWR; #45000-476) supplemented with 15% FBS (ATCC; #30-2020). The bulk of the ovary was severed from the oviduct, fully developed eggs were crudely removed, and individual ovarioles were roughly separated from each other. Each sample was then transferred to a sterile microfuge tube, and the media was aspirated. The remaining ovaries of each sample were then flash-frozen in liquid nitrogen and stored at –80°C prior to library preparation.

Total RNA libraries were made using the NEBNext Ultra II Directional RNA library Prep Kit for Illumina using the established protocol for use with NEBNext Poly(A) mRNA Magnetic Isolation Module (NEB#E7490). We used Rapid Run OBCG single-read 50 bp on the Illumina Hi-Seq 2500 system to sequence these libraries with two technical replicates for each timepoint and genotype. Sequenced reads were demultiplexed, and the indexes were removed using CASAVA v1.8.2 (Illumina).

## Single-cell RNA-sequencing sample preparation

$tj^{TS}>lgl^{RNAi}$ flies were shifted to 29°C, permissive temperature for the activation of transgenic expression of short-hairpins targeting the *l(2)gl* mRNA for functional knockdown in every cell of the follicular lineage. Ovaries from 50 such adult female flies were dissected 72 hr (3 days) later in complete medium and were then transferred to a tube containing 300 µl EBSS (TF, #78154; no calcium, magnesium, and phenol red) and were gently mixed for 2 min. After allowing the tissue to settle, EBSS was removed and individual cells were dissociated from the tissue in 100 µL Papain (Worthington, #LK03178; 50 U/mL in EBSS and heat-activated at 37°C for 15 min before use) for 30 min. The suspension was mechanically dissociated every 3 min by gentle pipetting up and down. To quench the digestion, 500 µL complete medium was subsequently added to the dissociated cells. The suspension was then passed through a 40 µL sterile cell strainer and centrifuged for 10 min at 700 RCF. The supernatant was then removed and 100 µL of complete medium was added. Viability and concentration of cells were assayed using propidium iodide staining, and manual cell count estimates were made using a hemocytometer to adjust input cell concentration for scRNA-seq library preparation.

In contrast to the 24 hr and 96 hr timepoints chosen for whole-tissue RNA-seq experiments, we chose 72 hr as the timepoint to construct the scRNA-seq dataset. This was chosen to mitigate the effects of increased germline cell death at 96 hr, which, at 96 hr, would produce significant debris that could contaminate the ovarian transcriptome by means of the ambient RNA within droplets. Additionally, widespread multilayering is first observed after 72 hr of Lgl-KD induction, which is significantly enhanced at 96 hr. We argued that whole-tissue RNA-seq analysis would only be able to capture statistically significant, multilayer-specific differential gene expression when the multilayers are significantly enhanced. However, given that gene expression is resolved for individual cells in scRNA-seq, preparing the dataset for cells derived from the 72 hr timepoint would be sufficient.

## 10X Genomics library preparation and sequencing

Single-cell libraries were prepared using the Single Cell 3′ Library & Gel Bead Kit v2 and Chip Kit (10X Genomics; PN-120237) according to the recommended 10X Genomics protocol, and the single-cell suspension was subsequently loaded on to the Chromium Controller (10X Genomics). Library quantification assays and quality check analysis were performed using the 2100 Bioanalyzer instrument (Agilent Technologies). The library samples were then diluted to the final concentration of 10 nM and loaded onto two lanes of the NovaSeq 6000 (Illumina) instrument flow cell for a 100-cycle sequencing run. Reads were demultiplexed the same way as described in bulk RNA-sequencing sample preparation and sequencing, and fastq files were generated for each library.

## Statistics and reproducibility

All statistical analyses of data and generation of graphs were performed in R. Boxes in box plots show the median and interquartile range; lines show the range of values within 1.5× of the interquartile range. Student's *t*-test has been used to perform statistical analyses. All phenotypic observations were independently assessed by two individuals, where one was blinded to the experimental goal (but aware of the experimental conditions). All images are representative of at least three independent experiments, except the *Act >>UAS-lgl^{RNAi}*, *UAS-GFP × UAS-CncC* experiment (**Figure 5D**) that was only repeated twice due to time constraints. Each experiment involves the dissection of a minimum of

25 flies per genetic cross, and multilayering of only stage 8–9 (midoogenesis) egg chambers in intact ovarioles was assessed for quantification. The genotypes for each experiment are clearly mentioned in each figure.

The scRNA-seq analysis of the $w^{1118}$ ovarian cells in this study reused the sequences generated for the published $w^{1118}$ ovarian cell atlas (*Jevitt et al., 2020*), which were obtained from the SRA database (SRX7814226). The number of cells passing the filtering criteria of Cell Ranger increased significantly from the previously reported number of 12,671 cells with 28,995 mean reads per cell to 24,144 cells with 17,095 mean reads per cell, with no change in the total number of sequenced reads (429,855,892). This change was a result of remapping of sequences to the top-level dm6 reference genome, which was different from the reference genome used in the Jevitt et al. study. This sequence realignment improved the genome alignment of the reads, and unlike the Jevitt et al. approach, manual curating was not performed on the dataset. The analytical pipeline in this study does not change the primary findings of the original paper as the markers described for each cell types remain unchanged.

## Whole-tissue RNA-sequencing analysis

Quality control on the sequences was performed using FastQC v0.11.9. Reads were mapped to the *Drosophila melanogaster* Release-6 (BDGP6.28.dna.toplevel) reference genome using STAR v2.7.0a (*Dobin et al., 2013*). BAM files resulting from STAR were sorted using the featureCounts program (http://bioinf.wehi.edu.au/featureCounts/) of the Subread package v2.0.1. Low-abundance counts of <1 counts per million (CPM) were removed ,and the downstream processing was performed using the edgeR pipeline 3.28.1 (*Robinson et al., 2009*) in R for the PCA of the two-sample experiment with two replicates each.

## Single-cell RNA-sequencing data processing

### Cell Ranger and Velocyto

Raw fastq sequencing files from each of the 10X Genomics Chromium single-cell 3' RNA-seq libraries were processed using Cell Ranger (version 3.0.0). The reference index for Cell Ranger was first built using the *Drosophila melanogaster* Release-6 (BDGP6.28.dna.toplevel) reference genome. The Cell Ranger count pipeline for alignment, filtering, barcode, and UMI counting was used to generate the multidimensional feature barcode matrix for all samples. The Cell Ranger-derived bam file for $tj^{TS}$>$lgl^{RNAi}$ sample was further processed using Velocyto CLI (default parameters) for the annotations of unspliced and spliced reads (*La Manno et al., 2018*).

### Seurat

The R package Seurat v3.0 (*Stuart et al., 2019*) is a universally popular software program to perform single-cell RNA-seq data preprocessing, dimensionality reduction, cell clustering analyses, differential gene expression analysis, and for general dataset handling. Since the detailed steps of data analyses in Seurat are explained on their website (https://satijalab.org/seurat/), we only describe the schematic workflow applied in this study. Each sample was filtered for low-quality cells by setting sample-specific thresholds for UMIs, gene counts, and mitochondrial gene expression. We used 1800 genes per cell as the upper threshold and 650 genes as the lower threshold for the experimental control sample and 1800 genes per cell and 600 genes as the upper and lower thresholds (respectively) for the $tj^{TS}$>$lgl^{RNAi}$ sample. Then only the cells having under 12,000 (experimental control) and 15,000 ($tj^{TS}$>$lgl^{RNAi}$) UMI counts were selected. Finally, an upper limit for the mitochondrial gene expression was applied to limit the selection of dying cells by filtering out cells expressing more than 10% (experimental control) and 4.5% ($tj^{TS}$>$lgl^{RNAi}$) of genes whose symbols begin with 'mt:' that is indicative of mitochondrial genes. As a result, 19,986 cells are finally obtained for the experimental control and 16,060 cells for $tj^{TS}$>$lgl^{RNAi}$ sample and these cells were subjected to downstream analyses. After the cells were embedded on lower UMAP dimensions following the suggested parameters for dimensionality reduction, the follicle cell lineage (epithelial cell type in the ovary) was further isolated from both the experimental control and $tj^{TS}$>$lgl^{RNAi}$ sample by removing irrelevant cell types using markers described in *Jevitt et al., 2020*.

For the integration of $w^{1118}$ and $tj^{TS}$>$lgl^{RNAi}$ follicle cell lineages, cells were aligned using the functions *FindIntegrationAnchors()* and *IntegrateData()* with 2000 genes for anchor finding and 50 dimensions for the canonical correlation analysis (CCA). UMI counts of the 'integrated' assay were normalized,

log-transformed, and scaled, following which the cells were clustered using 60 PCs and a resolution parameter of *1* across the two datasets. Cells were finally embedded on lower UMAP dimensions, using 100 neighboring points and a minimum distance of 0.5 for local approximations of manifold structure. The integrated dataset was carefully processed with an enforced awareness of the selected assay for downstream analyses. We used Seurat-integrated ALRA imputation (*Linderman et al., 2018*) on the sparse count matrix in the 'spliced' assay to recover missing values for gene count (default settings). The imputed matrix is stored in the 'alra' assay and was used only to generate gene enrichment plots. Gene enrichment plots for scRNA-seq datasets were scaled within the range of 5th and 95th quartiles to enhance visualization of cluster-specific marker enrichment. To identify differential markers, the analysis was restricted to the 'spliced' assay.

## Postprocessing of single-cell RNA-sequencing data

### scVelo

The loom file generated by Velocyto was used as an input in Seurat-based analysis. Subsetted $tj^{TS}>lgl^{RNAi}$ cells were further passed for lineage inference by running the scVelo (*Bergen et al., 2020*) pipeline on them. Without additionally processing the dataset, we directly estimated the underlying RNA velocity on the Seurat-determined cluster identities. We ran the dynamic model to learn the full transcriptional dynamics of the splicing kinetics in these cells. Velocity pseudotime, underlying latent time, and terminal end points were determined for the selected cells using default parameters. Cluster-specific genes that drive pronounced dynamic behavior were detected, and their phase portraits were generated where individual cells are colored according to their subcluster identities in cluster 7.

In this study, we have highlighted RNA velocity-derived interpretations that strictly agree with the other analytical perspectives pursued in this study. We applied scVelo to obtain information on the underlying lineage for (1) all unique Lgl-KD clusters and (2) cluster 7 cells. The cells of the unique Lgl-KD clusters represent a mixed population of mitotic, postmitotic, border follicle cells, and dying germline cell-associating cells that depict inconsistent transcriptional lineages. In this group of cells, the true developmental end point of the observed Lgl-KD lineage is cluster 8 (germline cell death occurs at the end of Lgl-KD follicular development), which likely consists of a mixed population of cells from the lateral epithelia as well as the multilayered epithelia, all responding to germline cell death. Indeed, certain sections of cluster 7 appear more similar to cluster 8 and others seem comparable to that of cluster 13. These observations underscore our conclusions that the unique Lgl-KD clusters exhibit distinguishable gene expression, representing different cell states. For cluster 7, the state of transcriptomic heterogeneity is what defines its unique state of gene expression and we have assessed this heterogeneity by specifically subsetting those cells.

### Regulon analysis using SCENIC

Regulon activity was assessed using the R package SCENIC v1.1.2-2 (*Aibar et al., 2017*). Expression matrix from the 'spliced' assay in each cell along with the corresponding Seurat metadata were used as input for SCENIC. The gene sets forming individual regulons were identified using the cisTarget v8 motif collection dm6-5kb-upstream-full-tx-11species.mc8nr. Overlapping regulon modules were identified based on the CSI, a context-dependent measure for identifying specific associating partners. CSI matrix was generated using the open-source R package scFunctions (https://rdrr.io/github/FloWuenne/scFunctions/).

## Acknowledgements

We thank Gengqiang Xie, Sumei Zhang, Roger Mercer, Yanming Yang, Cynthia Vied, Amber Brown, and Brian Washburn (at FSU Biological Science Sequencing Core) for their assistance in library preparation and sequencing. 10X Chromium controller and other essential hardware were provided by the FSU College of Medicine Translational Science Laboratory. We deeply appreciate feedback from all the members of Deng lab. Special thanks to Norbert Perrimon, Hugo Bellen, Jin Jiang, Dirk Bohmann, Jennifer Becker, and Matthew Rand, and the Gene Disruption Project for their contributions to generating and providing the transgenic lines used in this study.

## Additional information

### Funding
No external funding was received for this work.

### Author contributions
Deeptiman Chatterjee, Conceptualization, Data curation, Formal analysis, Validation, Investigation, Visualization, Methodology, Writing – original draft, Writing – review and editing; Caique Almeida Machado Costa, Validation, Visualization; Xian-Feng Wang, Investigation, Methodology, Writing – review and editing; Allison Jevitt, Data curation, Investigation, Methodology; Yi-Chun Huang, Resources, Validation; Wu-Min Deng, Conceptualization, Supervision, Funding acquisition, Project administration, Writing – review and editing

### Author ORCIDs
Deeptiman Chatterjee (iD) http://orcid.org/0000-0003-2752-4640
Wu-Min Deng (iD) http://orcid.org/0000-0002-9098-9769

### Decision letter and Author response
Decision letter https://doi.org/10.7554/eLife.80956.sa1
Author response https://doi.org/10.7554/eLife.80956.sa2

## Additional files

### Supplementary files
• MDAR checklist

• Source data 1. Enriched transcription factor binding motifs detected in genes expressed in the cells of cluster 7.

• Source data 2. Genes showing dynamic behavior of transcription driving the underlying RNA velocity of cluster 7 cells.

### Data availability
Both raw and processed sequencing data is available at GSE175435. Code necessary to replicate the main findings of this study is available at https://github.com/chatterjee89/eLife2022-11-e80956, (copy archived at swh:1:rev:b07579bd187bf32d99c5a2bf4836d0c2e9122e2d).

The following dataset was generated:

| Author(s) | Year | Dataset title | Dataset URL | Database and Identifier |
|---|---|---|---|---|
| Chatterjee D | 2021 | Data from: Single-cell transcriptomics uncovers Keap1-driven collective invasion upon epithelial polarity loss | https://www.ncbi.nlm.nih.gov/geo/query/acc.cgi?acc=GSE175435 | NCBI Gene Expression Omnibus, GSE175435 |

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
