## [Editor Report]

This work demonstrates the power of single-cell omics and imaging analyses to identify cell types and factors playing a role in the disruption of polarity, a process relevant to epithelial cancers. The authors' claims are well supported by the data and analyses. Overall, this work is viewed as an important contribution to cell biologists who work on the epithelial morphogenesis or tumorigenesis.

---

## [Decision Letter]

**Decision letter after peer review:**

Thank you for submitting your article "Single-cell transcriptomics identifies regulation of invasive behavior in *Drosophila* follicle cells with polarity loss" for consideration by *eLife*. Your article has been reviewed by 3 peer reviewers, and the evaluation has been overseen by a Reviewing Editor and David James as the Senior Editor. The following individuals involved in the review of your submission have agreed to reveal their identity: Mayu Inaba (Reviewer #1); Stefan Bekiranov (Reviewer #2); José M Martín-Durán (Reviewer #3).

Essential revisions:

1) Provide higher magnification images.

2) Determine (or predict) cells of each cluster, especially for some key clusters (including 7_3) within actual tissue by co-staining of several identified genes.

3) Measure Nrf2 protein levels Nrf2 protein levels in flies with and without Lgl-KD with various manipulations of Keap1 including control, KD, and OE.

4) Review major findings that show that Lgl is a tumor suppressor as is its human homologue Hugl-1 as well as making a stronger case that studying Lgl-KD in *Drosophila* is relevant for tumorigenesis and EMT. This information should be added to the discussion.

*Reviewer #1 (Recommendations for the authors):*

As I noted in public review, I would suggest authors to improve images now these are low magnification and difficult to see individual cells. Can authors define cells of each cluster, especially for some key clusters within actual tissue by co-staining of several identified genes?

*Reviewer #2 (Recommendations for the authors):*

The authors should address the weaknesses stated in the public review. They are sufficiently detailed and the relevant analyses, experiments, reframing and clarifications are effectively suggested there.

*Reviewer #3 (Recommendations for the authors):*

I do not have any major concerns on the manuscript. I think the analyses (computation and experimental) are done to high standards and the experimental validation of the single cell transcriptomic analyses is crucial to provide a clear understanding of the transcriptional changes brought by Lgl-KD. However, the authors might want to clarify the timings used for scRNA-seq and bulk RNA-seq. Why were 72h-induction originally chosen and selected for scRNA-seq? I do not think it is needed, but why did not the authors include 72h in the whole-tissue RNA-seq samples?

---

## [Author Response]

Essential revisions:1) Provide higher magnification images.

In response to the reviewer’s comments, we have modified the images in several figures. The revised manuscript has entirely new (or improved versions of) image panels in figure 5. In figure 1A, the focus is the entire ovariole and therefore, we have only highlighted the enrichment of Hnt and pH3 antibody staining separately for a subsetted region of interest (ROI). The ROI panels are included within the larger image itself. For figure 6, we have converted the LUTs of panels showing distinct channels for RFP and Shg/Arm antibody stainings to grayscale.

2) Determine (or predict) cells of each cluster, especially for some key clusters (including 7_3) within actual tissue by co-staining of several identified genes.

Unlike other studies where different clusters exhibit unique markers, the clusters derived from our integrated analysis do not exhibit mutually exclusive gene expression (with the exception of drpr/crq+ cluster 8). This is especially true for cluster 7, where the sub-clusters represent transitioning cell states and not distinguishable cell types. As is also evident from Figure 2B, the unique Lgl-KD specific clusters continue to show the enrichment of several stage-specific markers (e.g., *peb*, *mirr*, *brk*, *mid* and *slbo*) that are specific to the *wild-type* (*w*^*1118*^) follicle cells at corresponding developmental stages during midoogenesis. This result suggests that even though the Lgl-KD cells transcriptomically deviate from their corresponding cells of origin to form their own clusters, they continue to express several markers that show gene-expression overlap with normal follicle cells, thereby exacerbating the problem of identifying these cells using differentially-enriched markers.

To determine how the unique clusters of Lgl-KD distinguish from the *w^1118^* cells, we identified an upregulation of *JNK* signaling, Keap1-Nrf2 signaling as well as UPR-stress signaling, which are likely all related to each other. Puckered (puc) expression and pJNK staining has been detected in the egg chambers with either multilayers or apoptotic germline cells – however, *JNK* activity is not distinguishable between cluster 7 and cluster 8, as it is activated in both tumorigenic conditions as well as during apoptotic-cell clearance. We have validated the expression of Drpr (which is expressed in the Lgl-KD follicle cells surrounding the dying germline cells, as well as in the *w^1118^* follicle cells surrounding the dying nurse cells at later stages of normal development) and Mmp1 expression (which is expressed downstream of *JNK* signaling and only at later stages of terminated egg-chamber development containing when Lgl-KD is induced in follicle cells). Besides these 2 markers, we have also validated the presence of Keap1-Nrf2 signaling activity in the multilayered cells (Figure 4G). In the revised manuscript, we have provided further validation of UPR stress signaling activation by using the Xbp1-GFP sensor in the Lgl-KD egg chambers. We found Xbp1-GFP expression in all follicle cells of egg chambers that contain apoptotic follicle cells, with their earliest expression in the multilayered cells. Additionally, we validated Thor (the eukaryotic translation initiation factor 4E binding protein) expression in cluster 7 (mild expression) and cluster 8 (high expression). Images for pJNK, puc, Xbp1-GFP and Thor have been added to the Supplementary figures.

Among the markers enriched in the sub-groups of cluster 7 (Figure 4D), one can see that the differential markers enriched in each group is not so “differential” after all – as each marker shows some level of expression in almost all cluster-7 cells with only relative differences in their scaled expression. This is evident from GstD2 expression, which is highest in cluster 7_3 (Figure 4D) but is detected as a (relatively) differential marker of cluster 7 among the unique Lgl-KD clusters (Figure 3C) as well as in the integrated analysis with *w^1118^* clusters (Supplementary Figure 3B). In summary, multilayering/apically-invading Lgl-KD cells do not exhibit distinguishable gene-expression changes from their cells of origin, with the exception of cytoskeletal changes, *JNK* signaling and stress signaling – all 3 of which have been individually validated at the level of both mRNA expression as well as protein localization. Additional validation of the functional relevance of Keap1-Nrf2 pathway have been achieved via genetic-epistasis experiments. These clusters therefore most likely represent separable cell *states* and not distinguishable cell *types*. This rationale has been included in the revised manuscript.

3) Measure Nrf2 protein levels Nrf2 protein levels in flies with and without Lgl-KD with various manipulations of Keap1 including control, KD, and OE.

In the current manuscript, we have measured mRNA levels of Nrf2 (Cnc) as well as that of GstD2, the bona fide target of the Keap1-Nrf2 oxidative-stress signaling pathway in the different genetic backgrounds involving Lgl-KD, Keap1-OE and Keap1-KD as well as the experimental control. These assays do not represent expression levels within individual cells from the multilayers, as the samples are derived from whole ovaries. Given the lack of distinguishable markers, we cannot sort single cells from within the multilayers and so, detecting protein levels would also suffer from similar shortcomings. GstD-lacZ expression being sporadic and transient in individual cells (as every cell responds to oxidative stress differently), it cannot be used as a quantifiable measure of pathway activation. However, its expression is differentially observed within the multilayers, leading us to only conclude that the pathway is indeed active in the invasive multilayered cells.

From the results discussed in the previous version of the manuscript, it was unclear how the mutually-antagonizing relationship between Keap1 and Nrf2 (that is essential to their antioxidant response) effects the observed phenotype. In the revised manuscript, we have performed additional experiments to comprehensively conclude that the phenotype is not dependent on the relationship between Keap1 and Nrf2. Along with the evidence that other oxidative-stress response genes (Nox, Duox, Sod1 and Cat) are not active in cluster 7 and that ROS is not detected in these cells (no significant DHE staining), we believe that Keap1-Nrf2 mediated oxidative-stress response is likely not regulating the invasive phenotype directly. Knocking down both Keap1 and Cnc decreases multilayer growth and overexpressing them individually promotes Lgl-KD invasion. In light of the genetic epistasis experiments (that have made the assessment of Keap1 and Nrf2 expression levels irrelevant) and the abovementioned evidence (as well as the aforementioned technical difficulties), measuring Nrf2 protein levels won’t be able to add any significance to the study.

4) Review major findings that show that Lgl is a tumor suppressor as is its human homologue Hugl-1 as well as making a stronger case that studying Lgl-KD in *Drosophila* is relevant for tumorigenesis and EMT. This information should be added to the discussion.

We have addressed this concern by rewriting the Discussion section. In the first paragraph, we have discussed how inducing polarity loss via Lgl knockdown is relevant to human cancers. We have also added new paragraphs discussing Keap1’s role in cytoskeletal regulation and how they may relate to the different aspects of metastasis.

Reviewer #3 (Recommendations for the authors):I do not have any major concerns on the manuscript. I think the analyses (computation and experimental) are done to high standards and the experimental validation of the single cell transcriptomic analyses is crucial to provide a clear understanding of the transcriptional changes brought by Lgl-KD. However, the authors might want to clarify the timings used for scRNA-seq and bulk RNA-seq. Why were 72h-induction originally chosen and selected for scRNA-seq? I do not think it is needed, but why did not the authors include 72h in the whole-tissue RNA-seq samples?

We thank the reviewer for their words of affirmation, and hope that they will also appreciate the revisions that now make this story have an even sharper message. The following paragraph has been added to the Methods section (within the subsection “Single-Cell RNA-Sequencing Sample Preparation”) to address this issue:

“In contrast to the 24h and 96h time-points chosen for whole-tissue RNA-Seq experiments, we chose 72h as the time-point to construct the scRNA-Seq dataset. This was chosen to mitigate the effects of increased germline cell-death at 96h which, at 96h, would produce significant debris that could contaminate the ovarian transcriptome by means of the ambient RNA within droplets. Additionally, wide-spread multilayering is first observed after 72h of Lgl-KD induction, which is significantly enhanced at 96h. We argued that whole-tissue RNA-Seq analysis would only be able to capture statistically significant, multilayer-specific differential gene expression when the multilayers are significantly enhanced. However, given that gene expression is resolved for individual cells in scRNA-Seq, preparing the dataset for cells derived from the 72h time-point would be sufficient.”